# A metal-free photocatalyst for highly efficient hydrogen peroxide photoproduction in real seawater

Qingyao Wu[1], Jingjing Cao[1], Xiao Wang[1], Yan Liu[1], Yajie Zhao[1], Hui Wang[1], Yang Liu[1✉], Hui Huang[1], Fan Liao[1], Mingwang Shao[1✉] & Zhenghui Kang [1,2✉]

Artificial photosynthesis of $H_2O_2$ from $H_2O$ and $O_2$, as a spotless method, has aroused widespread interest. Up to date, most photocatalysts still suffer from serious salt-deactivated effects with huge consumption of photogenerated charges, which severely limit their wide application. Herein, by using a phenolic condensation approach, carbon dots, organic dye molecule procyanidins and 4-methoxybenzaldehyde are composed into a metal-free photocatalyst for the photosynthetic production of $H_2O_2$ in seawater. This catalyst exhibits high photocatalytic ability to produce $H_2O_2$ with the yield of 1776 μmol g$^{-1}$h$^{-1}$ ($\lambda \geq 420$ nm; 34.8 mW cm$^{-2}$) in real seawater, about 4.8 times higher than the pure polymer. Combining with in-situ photoelectrochemical and transient photovoltage analysis, the active site and the catalytic mechanism of this composite catalyst in seawater are also clearly clarified. This work opens up an avenue for a highly efficient and practical, available catalyst for $H_2O_2$ photoproduction in real seawater.

[1] Institute of Functional Nano and Soft Materials Laboratory (FUNSOM), Jiangsu Key Laboratory for Carbon-Based Functional Materials & Devices, Soochow University, 215123 Suzhou, PR China. [2] Macao Institute of Materials Science and Engineering, Macau University of Science and Technology, Taipa, 999078 Macau SAR, China. ✉email: yangl@suda.edu.cn; mwshao@suda.edu.cn; zhkang@suda.edu.cn

Hydrogen peroxide ($H_2O_2$) has aroused widespread interest as a high-energy oxidant for the potentially replacement of fossil fuel because of its widely application in the field of medicine, chemical industry, and environmental management[1–3]. The traditional approaches for $H_2O_2$ production, such as anthraquinone method and electrochemical synthesis, result in huge energy consumption and high toxicity by-products[4,5]. Artificial photosynthesis method, namely the reaction of oxygen and water driven by sunlight, is recognized as the most promising, spotless method for the $H_2O_2$ green fabrication[6–8]. Given the abundance of seawater, the utilization of seawater to photo-produce $H_2O_2$ can not only mitigate the problem of fresh water resource shortage but also greatly reduce the cost of reaction system process[9,10]. Although more and more materials are being used in photocatalytic synthesis of $H_2O_2$, most of them are still limited by such factors as poor light absorption[11,12], mismatched band gap[13,14], low electron transfer rate[15], and essential sacrificial agents[16,17]. In particular, salt can deactivate the catalysts or consume photogenerated carriers, leading to undesirable side reactions on the surface of the catalyst, which severely limits the industrialization process of $H_2O_2$ production by sunlight[18,19].

In photocatalytic field, carbon dots (CDs) have been proved to be the co-catalytic active site and/or the good electron acceptor/donor material, and show unique ability to improve the catalytic efficiency in the photocatalytic system[20–23]. Herein, as shown in Fig. 1, we report a phenolic condensation approach, in which CDs, organic dye molecule procyanidins, and 4-methoxybenzaldehyde are composed into metal-free photocatalyst (PM-CDs-x) for the photocatalytic production of $H_2O_2$ in real seawater. Notably, we propose and prove the electron sink effect of CDs, and this electron sink effect would grow with the addition of metal cations. So, the catalyst can extract more electrons under light excitation, effectively hindering the electron–hole recombination. It turns out that this ideal composite catalyst exhibits the expected photocatalytic capacity to generate $H_2O_2$ in seawater, and the catalytic activity of the optimal composite photocatalyst is approximately 4.8 times high than the pure polymer photocatalyst in seawater. In particular, the maximum yield of $H_2O_2$ for the optimal catalyst PM-CDs-30 is 1776 $\mu mol\,g^{-1}\,h^{-1}$ in seawater, the apparent quantum yield (AQY) is as high as 0.54% at 630 nm in seawater, and the solar-to-chemical conversion (SCC) efficiency can reach 0.21% in seawater. Combining with $H_2O_2$ generation rates, in situ photoelectrochemical and transient photovoltage (TPV) tests, the role of CDs in the composite catalyst PM-CDs-30, during the photocatalytic reaction process, was established, and the active site and the working mechanism of the composite catalyst in seawater were also clarified. Moreover, as suggested in a recent article[24], the optimal reaction conditions of the catalyst were investigated through experiments and thermodynamic–kinetic model. Our results open up an avenue for seawater utilization, catalyst design, and $H_2O_2$ generation.

## Results

**Characterizations of CDs.** Firstly, the structure and photoelectrochemical properties of CDs were studied. Figure 2a shows the transmission electron microscopy (TEM) image and particle size distribution taken from monodispersed CDs, making clear that CDs are spherical nanoparticles with a diameter of about 6–16 nm. A high-resolution TEM (HRTEM) image displayed in Fig. 2b exhibits a lattice spacing of 0.21 nm, which corresponds to (100) lattice planes of graphitic carbon[19]. The functional groups of CDs are indicated in the Fourier transform infrared (FT-IR) spectrum (inset in Fig. 2b), where the peaks located at 1632, 1397, and 1233 $cm^{-1}$ are assigned to the stretching vibration of C=O, –COO, and C–OH[25], respectively. Power X-ray diffraction (XRD) pattern of CDs in Fig. S1 shows two broad peaks at around 21° and 43° corresponding to the (002) and (100) planes of graphitic carbon.

The full X-ray photoelectron spectroscopy (XPS) spectrum in Fig. S2a demonstrates only C and O elements in CDs. The high-resolution spectrum of C 1s can be fitted for C–C/C=C, C–O, and C=O while the O 1s can be matched to $O_{surf}$ and $O_{ads}$[26,27]. Further structure analysis on CDs is shown in Fig. 2c, where the hydrion in the carboxyl group shows electropositivity ($\delta_1^+$) while the adjacent oxygen is electronegativity ($\delta_1^-$). When the metal salt ($M^+$) is added, the carboxyl group can be ionized, and the electronegativity ($\delta_2^-$) of oxygen in the carbon group increase, which will, in turn, increase the charges and electron sink barrier of CDs, and further prolongs the life of the electron. Density functional theory calculations have been conducted to understand the effect of ions on the electrostatic potentials of CDs with respect to the energy level of a vacuum. As shown in inset images of Fig. S3, CDs are modeled with one-dimensional graphene nanoribbons, the edge of which are functionalized with carboxyl (−COOH) and sodium carboxylate (−COONa). The computational details can be found in the supporting information. The work function of these graphene nanoribbons can be estimated by plane-averaged potentials. Figure S3 shows the plane-averaged potentials of nanoribbons are plotted along the direction that perpendicular to the surface. It is seen that the trapping of electrons is more profound in the graphene nanoribbon terminated with carboxyl groups. This is in good agreement with our predictions.

The TPV experiments were carried out and shown in Fig. 2d, in which the TPV curve of CDs decays slowly and gently, indicating a slow combination of charges. Obviously, after the addition of NaCl, the intensity of photovoltage increased, which is mainly because that the enhanced electron-withdrawing property of the functional group by ionization increases electron extraction rate under the same excitation. Furthermore, due to the increased energy barrier of the electron sink, the "trapped" electrons surrounding adjacent oxygen cannot participate in the electron–hole recombination process in a short time. Therefore, CDs draw electrons from the

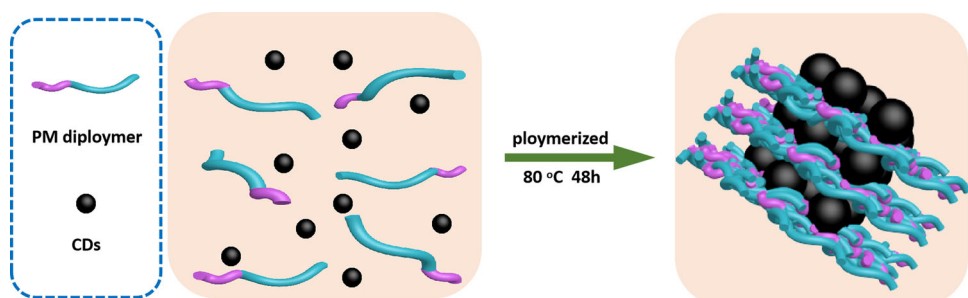

**Fig. 1 Synthesis process.** The synthesis process of organic polymer composite systems. PM dipolymer represents procyanidins-methoxybenzaldehyde diplymer.

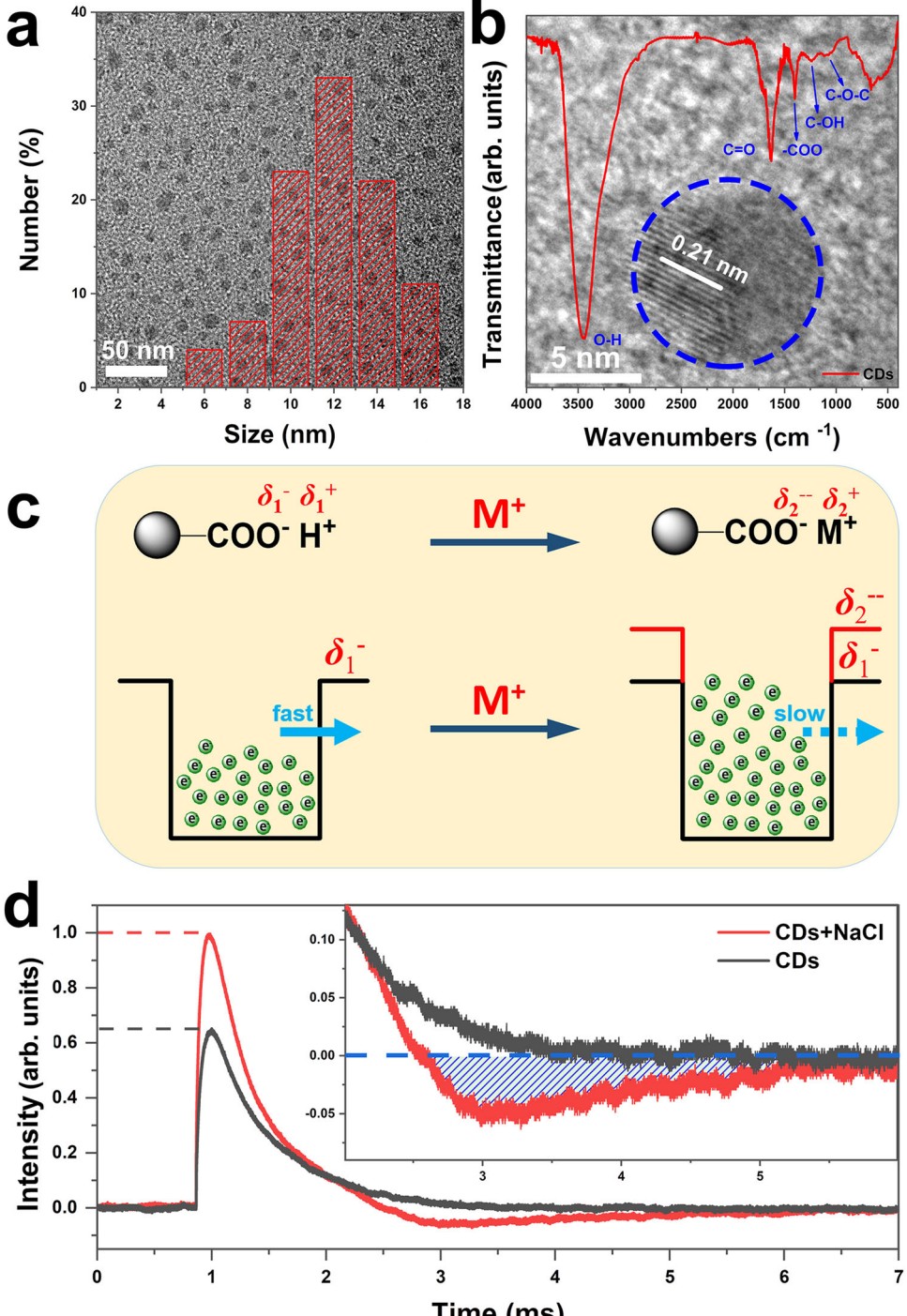

**Fig. 2 Structure and property of CDs. a** TEM image of CDs with the particle size distribution inserted. **b** HRTEM image of CDs with FT-IR spectrum inserted. **c** Schematic diagram of the electron sink model for CDs. The black boll represents a single CD, $M^+$ represents metal cations. **d** TPV curves of CDs powders before and after adding NaCl.

system, which causes negative signal region on the TPV curve (the red line in Fig. 2d). As time goes on, the "trapped" electrons slowly pass through the barrier and recombine with holes, eventually reaching equilibrium. The TPV tests of CDs mixed with other metal salt ions shown in Fig. S5 display the same phenomenon as that added with NaCl. Another important property in Fig. S6 reveals that CDs can perform electrocatalytic oxygen reduction reaction (ORR) by two-electron channel with no photoelectric

enhancement effect, suggesting that it can be used as the ORR site in the catalytic system. Thus, the functional groups (such as –OH, C=O, –COOH) on the surface of CDs not merely play an extraordinary role in the design and performance regulation of catalysts, but also effectively trap electrons through its electron sink effect in seawater to accelerate the electron transfer and prevent electron–hole recombination, thereby improving the photocatalytic activity of the catalyst in seawater.

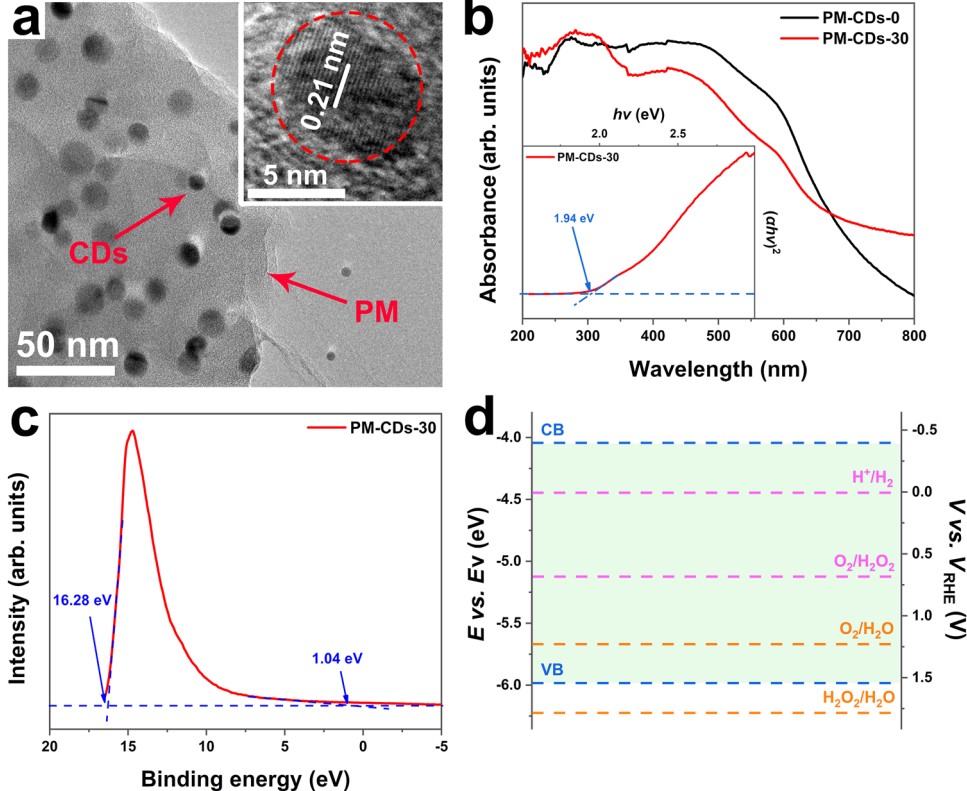

**Fig. 3 Structure and property of PM-CDs-30. a** TEM image and HRTEM image (inset), **b** UV–vis absorption spectra (inset: the corresponding Tauc plot), **c** UPS spectrum, and **d** band structure diagram of PM-CDs-30.

**Characterizations of composite photocatalysts.** PM-CDs-$x$ was prepared by adding CDs into the crosslinked polymerization of procyanidins-methoxybenzaldehyde (PM) dipolymer. Here, PM-CDs-30 is taken as an example to show the structure of the photocatalyst. Scanning electron microscope (SEM) image of PM-CDs-30 (Fig. S7a) and its partial enlarged view (Fig. S7b) indicate homogeneous micron-sized particles with flake accumulation, while the SEM image of PM-CDs-0 (Fig. S7c) shows nonuniform spherical particles. Further observation of the structure through TEM (Fig. 3a) shows that CDs with the size of 6–16 nm are uniformly distributed in the lamellar layer of the polymer PM-CDs-30. A HRTEM image inserted in Fig. 3a gives a lattice spacing of 0.21 nm for the single CD in PM-CDs-30, corresponding to the (100) lattice planes of graphitic carbon. In addition, a series of basic characterizations, such as the size distribution (Fig. S8), XRD patterns (Fig. S9a), FT-IR spectra (Fig. S9b), XPS spectra (Fig. S10), elemental analysis (Table S1), and specific surface area (Table S2), declare that PM-CDs-30 is an amorphous carbon structure composed of C, H, O elements with abundant functional groups.

The optical properties and band structure of PM-CDs-30 were examined by ultraviolet–visible (UV–vis) absorption spectroscopy and ultraviolet photoelectron spectroscopy (UPS) test. UV–vis adsorption spectra in Fig. 3b demonstrate that PM-CDs-30 shows well absorption performance in ultraviolet, visible, and even near-infrared region. It is worth noting that the addition of CDs enhances the absorption of photocatalyst at the wavelength greater than 670 nm. As the content of CDs increases, the color of the catalyst deepened (Fig. S11), and the absorption in the near-infrared region also gradually increases, which is more conducive to the utilization of sunlight (Fig. S12). The optical band gap of PM-CDs-30 can be calculated from the Tauc plot (inset in Fig. 3b) derived from the UV–vis absorption spectra.

The switched Tauc plot of $(\alpha h v)^{1/2}$ versus photon energy ($hv$) is obtained according to the formula introduced in the previous works[12], and the band gap energy ($E_g$) of PM-CDs-30 is determined to be 1.94 eV. Then, UPS is used to estimate the ionization potential (equal to the valence band energy ($E_{VB}$)) and investigate the band position of PM-CDs-30. According to Fig. 3c, by subtracting the width of the He I UPS spectrum from the excitation energy (21.22 eV), the $E_{VB}$ of PM-CDs-30 is determined to be 5.98 eV (vs. vacuum). Thus, the conduction band ($E_{CB}$) is calculated to be 4.04 eV (vs. vacuum). Visualizing, the energy level diagram of PM-CDs-30 is displayed in Fig. 3d, indicating that the energy level of the conduction band of PM-CDs-30 is higher than that of the ORR (0.68 V vs. RHE). Noting that the energy level of the valence band is located below the oxidation level for $O_2/H_2O$ (1.23 eV vs. RHE) but above than that for $H_2O_2/H_2O$ (1.78 eV vs. RHE). Thence, the organic polymer composite PM-CDs-30 with appropriate band position can realize the ORR and water oxidation reaction simultaneously.

**Photocatalytic activity of the composite systems PM-CDs-$x$.** The photocatalytic activities of the as-prepared photocatalysts were studied in pure water and real seawater at room temperature and pressure, and the results are summarized in Fig. 4a. Obviously, the pure polymer PM-CDs-0 generates $H_2O_2$ with a low production rate of 771 μmol g$^{-1}$ h$^{-1}$ in pure water. While, when the catalytic environment is changed to seawater, the activity of the catalyst reduced greatly, which is about 0.48 times that of pure water. This is mainly due to the large number of ionic components and impurities in seawater seriously affected the intrinsic structure and electron transport of photocatalysts[19,28,29]. Relatively speaking, PM-CDs-30 shows the best activity in both pure water and seawater solution among the

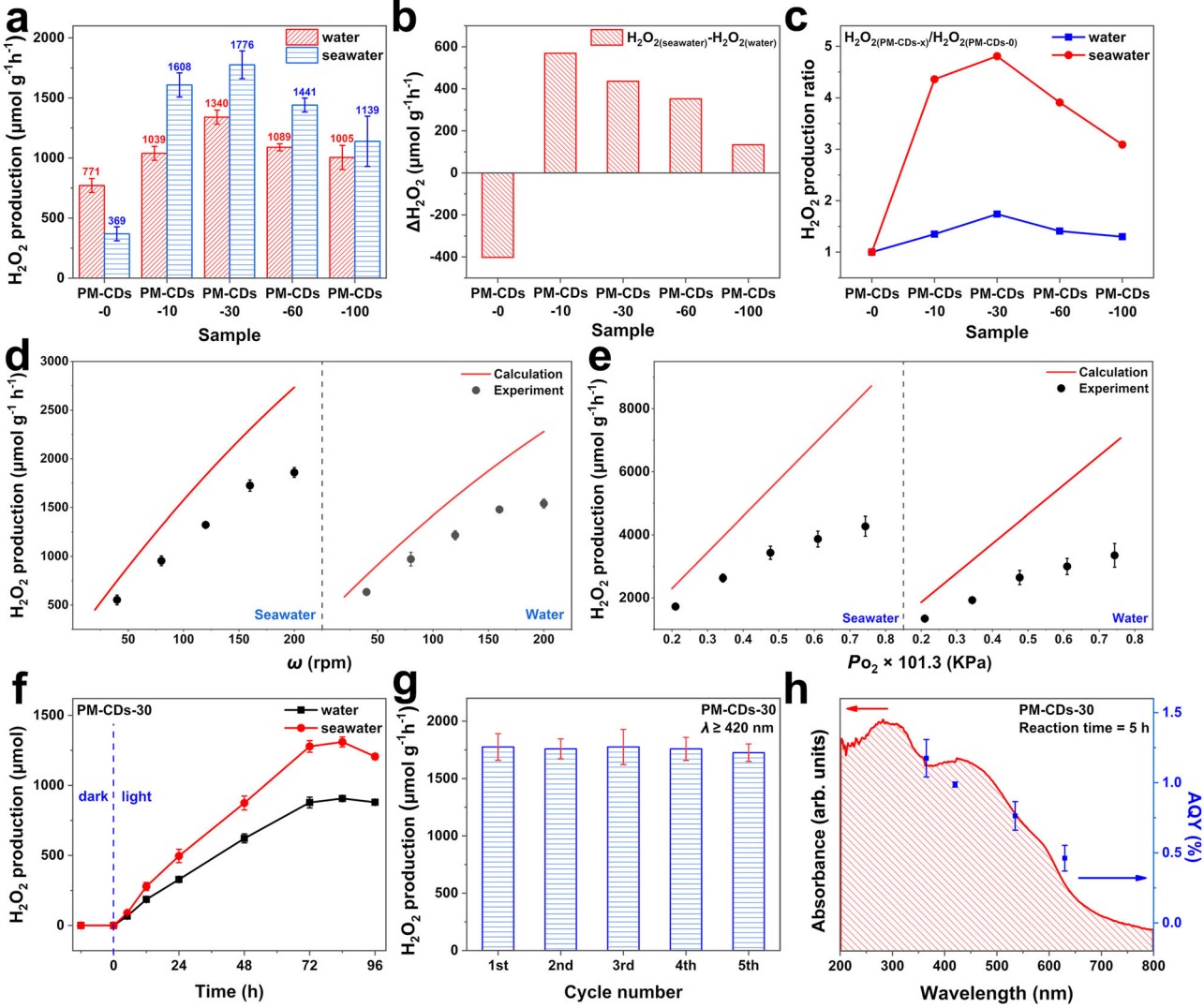

**Fig. 4 Catalytic performance of PM-CDs-30. a** Comparison of $H_2O_2$ production among the photocatalysts with different CDs contents in pure water and real seawater. **b** Comparison of hydrogen peroxide yield rate between seawater and water with different catalysts. **c** The hydrogen peroxide production ratio with different CDs contents to pure polymer catalyst PM-CDs-0. **d** Calculation and experimental on the dependence of $H_2O_2$ production rate of PM-CDs-30 and rotational speed in different medium (red line: calculation data; black point: experimental data). **e** Calculation and experimental on the dependence of $H_2O_2$ production rate of PM-CDs-30 and oxygen partial pressure in different medium (red line: calculation data; black point: experimental data). **f** Time course of $H_2O_2$ photoproduction by PM-CDs-30. **g** Stability of photoproduction of $H_2O_2$ by PM-CDs-30. **h** Wavelength-dependent AQY of oxygen reduction reaction by PM-CDs-30. The vertical error bars indicate the maximum and minimum values obtained; the dot represents the average value.

catalysts studied in this work, and its photocatalytic activity in seawater (1776 μmol g$^{-1}$ h$^{-1}$) is much higher than that in pure water (1340 μmol g$^{-1}$ h$^{-1}$). And the SCC efficiency of PM-CDs-30 can reach 0.21% in seawater. To further explore the photocatalytic activity of the catalysts in real seawater and pure water, the difference in hydrogen peroxide production rate between seawater and pure water is displayed in Fig. 4b. Surprisingly, the polymers with CDs added all show improved photocatalytic activity in seawater compared with pure water, and its photocatalytic activity in seawater is about 1.1–1.5 times than that of in pure water. Importantly, Fig. 4c clearly shows the effect of the added CDs on the photocatalytic performance of the composite catalyst, indicating that the addition of CDs can increase the catalytic activity about by up to 4.8 times compared with pure polymer PM-CDs-0 in real seawater. In summary, CDs are useful

for photocatalytic ORR, especially play an important role in the photocatalytic reaction under seawater condition. In terms of ORR photocatalytic activity, PM-CDs-30 composite was determined to be an optimal ratio of CDs in this composite.

The thermodynamic–kinetic model was performed to predict the optimum reaction conditions and maximum reaction rate of PM-CDs-30. Considering various factors that affect reactions rates during the photocatalytic reactions, the rate optimization is done with freely varying lattice parameters by using electrochemical methods and in situ TPV tests. The detailed calculation process is shown in the supporting information. Here, we consider the influence of water oxidation, oxygen reduction, and oxygen diffusion on the reaction rate, and obtained the relationship between oxygen partial pressure, rotational speed, and reaction rate. First, for oxygen evolution reaction (OER) in

seawater, the reaction rate $r_1$ can be obtained.

$$2H_2O + 4h^+ \xrightarrow{r_1} O_2 + 4H^+$$

$$r_1 = \lambda n_1 F k_{10} \exp\left(\frac{-E_1}{RT}\right) \exp\left(\frac{\alpha F(U_1 - U_{10})}{RT}\right) C_1^m = 8.97 \times 10^4 \tag{1}$$

where $\lambda$ is rate ratio of the half reactions; $n_1$ is the number of hole transfer ($n_1 = 4$); $F$ is the Faraday constant ($F = 96485$ C mol$^{-1}$); $k_{10}$ is the rate constant of OER; $E_1$ is the activation energy of OER; $R$ is the gas constant ($R = 8.314$ J mol$^{-1}$ K$^{-1}$); $T$ is the reaction temperature ($T = 298.15$ K); $\alpha$ is a constant ($\alpha = 0.5$); $U_1$ is a potential (vs. RHE) in the Tafel region; $U_{10}$ is the potential of water oxidation ($U_{10} = 1.23$ V); $C_1^m$ is the concentration of hydroxyl of water ($m = 1$). Then, for ORR in seawater, the reaction rate $r_2$ can be obtained.

$$O_2 + 2H_2O + 2e^- \xrightarrow{r_2} H_2O_2 + 2OH^-$$

$$r_2 = n_2 F k_{20} \exp\left(\frac{-E_2}{RT}\right) \exp\left(\frac{\alpha F(U_2 - U_{20})}{RT}\right) C_2^n$$

$$= 2.31 \times 10^{11} \times PO_2 \times \left(\frac{\omega}{1600}\right)^{0.5} (\omega > 0) \tag{2}$$

where $n_2$ is the number of hole transfer ($n_2 = 2$); $k_{20}$ is the rate constant of ORR; $E_2$ is the activation energy of ORR; $U_2$ is a potential (vs. RHE) in the Tafel region; $U_{20}$ is the potential of water oxidation ($U_{20} = 0.68$ V); $C_2^n$ is the partial pressure of oxygen ($n = 1$); $\omega$ is the rotation speed in the reaction system (r.p.m.); $PO_2$ is the partial pressure of oxygen in the reaction system. In particular, the rotational speed here is greater than zero. Next, the reaction rate ($r_3$) of oxygen diffusion in ORR in seawater is not negligible.

$$r_3 = J_{max}(60 \text{ r.p.m.}) \times \left(\frac{\omega}{60}\right)^{0.8} \times PO_2 = 5.24 \times 10^3 \times \left(\frac{\omega}{60}\right)^{0.8} \times PO_2 (\omega > 0) \tag{3}$$

By comparing all the reaction rates of the photocatalytic reaction under certain conditions, the rate-limiting step is obtained, that is, the reaction rate under this condition. Depending on the above calculation, the thermodynamic–kinetic model of PM-CDs-30 freely varying with rotational speed and oxygen partial pressure in seawater is displayed in Fig. S16a, indicating that the reaction rate increases with the increase of rotational speed, which is a gradual process. Meanwhile, within the calculation range, the reaction rate increases continuously with the improvement of oxygen partial pressure. It should be noted that, the rate of OER ($r_1$) has no effect on the production rate of $H_2O_2$ production. This is mainly attributed to the fact that ORR and OER are parallel reactions in this photocatalytic process, where the ORR process mainly relies on the dissolved oxygen in seawater from air.

Figure 4d shows the experimental and theoretical data on the relationship between catalytic activity and rotational speed in both seawater and pure water. Similar to the calculation, the yield rate of $H_2O_2$ increased with the increase of the rotational speed. The same situation is also applicable to Fig. 4e, the theoretical and experimental comparison of oxygen partial pressure and $H_2O_2$ yield, where the relationship between catalytic activity and oxygen partial pressure is consistent with the calculated results. It should be pointed out that experimental data obtained is lower than the calculated results because that the decomposition of $H_2O_2$ is not taken into account in the theoretical calculation. For comparison, the photocatalytic rates of catalyst in pure water were also calculated and verified. As shown in Fig. S16b, the variation trend of PM-CDs-30 reaction rate in water with rotation speed and oxygen partial pressure is similar to that in seawater, but the

maximum rate is far lower than that in seawater, which is consistent with the experimental results in Fig. 4a. As shown in Fig. 4d, e, the comparison of theoretical and experimental data in both seawater and pure water illustrate the accuracy and feasibility of the thermodynamic–kinetic model.

The variation trends of $H_2O_2$ generation over time are investigated in Fig. 4f. Specifically, no $H_2O_2$ is detected under the darkness in either pure water or seawater, but the yield of $H_2O_2$ is linear with time at the early stage of irradiation, suggesting that photocatalytic ORR is related to light. However, after 72 h reaction, the yield of $H_2O_2$ almost remains unchanged, and even shows a slight downward trend, which may be attributed to the decrease of catalytic activity and/or the decomposition of $H_2O_2$. In addition, the aggregation experiments in Fig. S19 suggest that the catalyst aggregate does not appear when the salt concentration in the range of seawater.

The re-usability of PM-CDs-30 has been studied via photocatalytic experiments using recovered samples. As shown in Fig. 4g, after five cycles of photocatalytic reaction, the $H_2O_2$ formation rate of PM-CDs-30 still remains at 1725 μmol g$^{-1}$ h$^{-1}$, indicating that PM-CDs-30 exhibits high stability in seawater. The excellent stability of PM-CDs-30 was further corroborated by electron microscopy and XPS analysis. The SEM image (Fig. S20a) shows no obvious changes in morphology, while the TEM image (Fig. 20b) indicates a slight aggregation of CDs after long times photocatalytic reaction. Moreover, there are many signal peaks of elements in the full XPS spectrum after the catalytic reaction (Fig. S21a), such as Na and Cl, compared with those before the catalytic reaction, which are mainly caused by the adsorbed ions and impurities from seawater. While, the C 1s and O 1s spectra of PM-CDs-30 show no obvious change after continuous photocatalytic reaction (Fig. S21b, S21c).

The AQY was measured to evaluate the light utilization efficiency of PM-CDs-30 under the illumination of monochromatic light. As the action spectrum shown in Fig. 4h, the AQY accords closely with the absorption spectrum of the photocatalyst, demonstrating that the band gap is excited to generate $H_2O_2$. According to the equation in the experimental section, the AQY of PM-CDs-30 can still reach 0.54% under the excitation light of 630 nm in seawater. Therefore, the catalyst can convert chemical energy well because the visible and near-infrared light account for the most of the solar energy.

**Proposed photocatalytic mechanism.** The electron chemical impedance spectroscopy (EIS) technique was used to probe the charge-transfer properties of the catalysts (PM-CDs-x, Fig. S22a). The smaller semicircle for PM-CDs-30 indicates a faster electron transfer than others, suggesting that PM-CDs-30 possess the best migration property[30]. Time-dependent photo-response curves of these samples are tested with intermittent visible light ($\lambda \geq$ 420 nm) at open-circuit voltage in seawater (Fig. S22b). It indicates that PM-CDs-30 exhibits the best photo-response property, while PM-CDs-0 shows the worst compared to others, strongly suggesting that PM-CDs-30 has superior charge generation property compared to pristine polymer PM-CDs-0[31]. A series of in situ TPV tests on PM-CDs-x (from PM-CDs-0 to PM-CDs-30) were carried out and shown in Fig. S22c. A closer observation reveals that the curve of the catalyst with a small amount of CDs content (PM-CDs-10) decays slower than the curve of catalyst without CDs (PM-CDs-0). As the increase of CDs content, the curves decay to a negative value region, which expands as the CDs content increases. All these results suggest that CDs promote charge transfer at the interface of polymer and CDs, and effectively improve the light absorption, charge carrier separation and conductivity of the catalyst. Also, CDs increase the electron sink

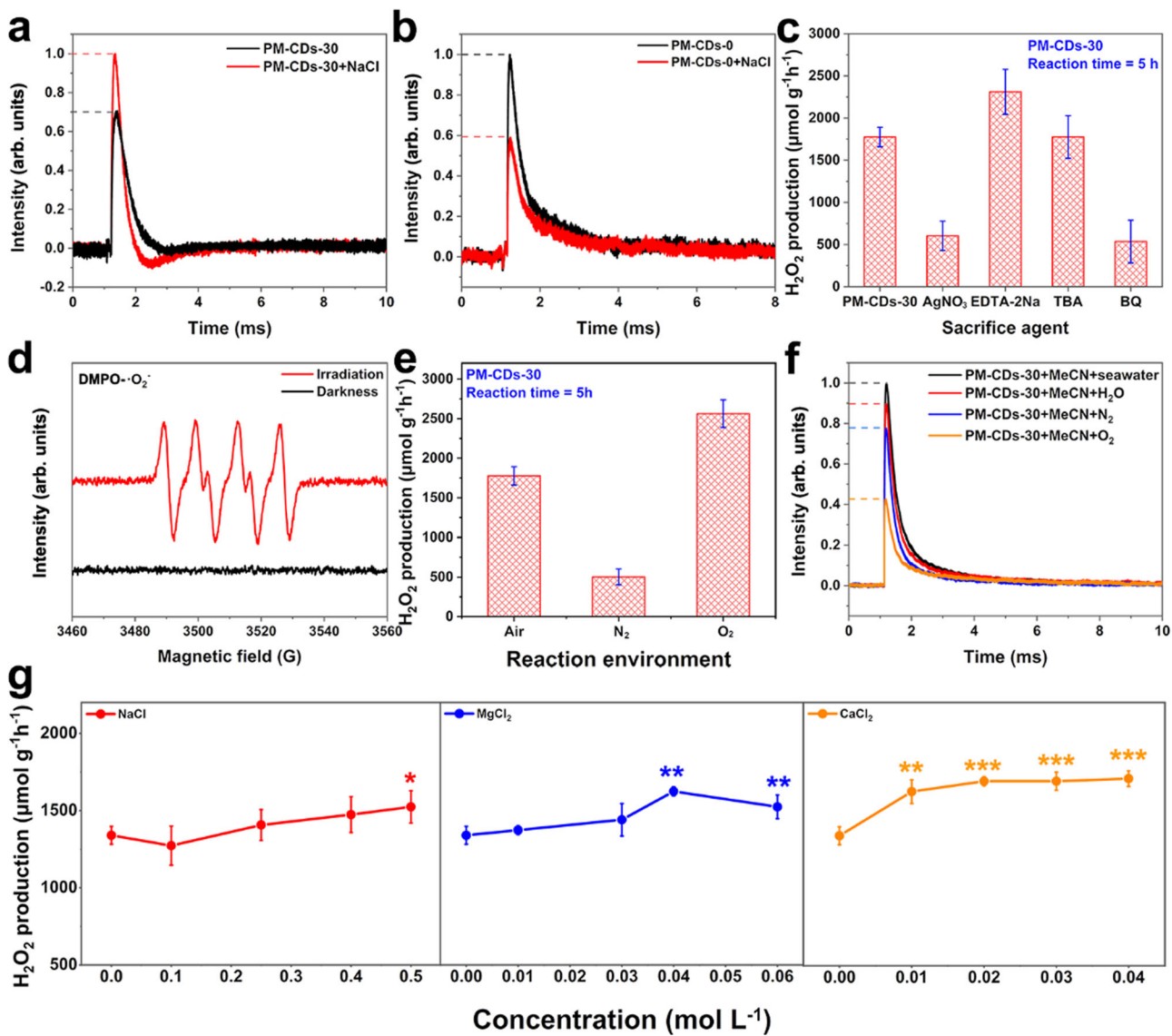

**Fig. 5 Photocatalytic mechanism of PM-CDs-30.** TPV curves of **a** PM-CDs-0, **b** PM-CDs-30 before and after adding NaCl. **c** The photocatalytic $H_2O_2$ production rates upon the addition of different sacrificial agents. **d** EPR diagrams of PM-CDs-30 under darkness and light. **e** The production rate of $H_2O_2$ by photocatalytic reaction in different gas environments. **f** TPV curves of PM-CDs-30 under different conditions. **g** $H_2O_2$ production rate varies with ion concentration in different salt solutions. *$P < 0.10$; **$P < 0.05$; ***$P < 0.01$. The vertical error bars indicate the maximum and minimum values obtained; the dot represents the average value.

barrier and electron trap ability of photoelectron and further inhibit the recombination of electron–hole pairs in PM-CDs-$x$.

Systematic TPV experiments of the catalytic systems with (PM-CDs-30) and without CDs (PM-CDs-0) were carried out to explore the mechanism of enhanced catalytic performance of salt ions and CDs. The catalyst (50 mg) and NaCl (0.003 mol) were dissolved and lyophilized. Then the uniformly mixed sample was excited under the same conditions as the original catalyst PM-CDs-0 and PM-CDs-30. As shown in Fig. 5a, the catalyst mixed with NaCl produced more charges than the original catalyst, and the negative region generated by charges recombination in attenuation was expanded. However, as a comparison, PM-CDs-0 (Fig. 5b) not only does not promote the generation of photoexcited charges, but also does not appear in the negative region during the decay time, indicating that the generation of negative region is mainly caused by the action of CDs in the polymer. The most likely reason for the negative region enlarging in the decay process is that the ionization of CDs by Na$^+$ increase

the electron sink effect to "trapped" electron transfer, thus attracting more electrons from the detection system. In the presence of CDs, the TPV curves of PM-CDs-30 mixed with other salt ions (Fig. S25), such as $MgCl_2$ and $CaCl_2$, also show similar phenomenon to NaCl. In addition, the difference significance test of statistics method was adopted in the final analysis, indicating that the cations can promote the production of hydrogen peroxide in seawater.

To understand the mechanism of photocatalytic $H_2O_2$ production over the PM-CDs-30 in seawater, by using AgNO$_3$, EDTA-2Na, *tert*-butyl alcohol (TBA), and benzoquinone (BQ) as electron ($e^-$), hole ($h^+$), hydroxyl radical (·OH), and superoxide radical (·$O_2^-$) scavengers, the active species trapping experiments were performed[4,31]. As shown in Fig. 5c, when AgNO$_3$ is added to the photocatalytic reaction system, the yield of $H_2O_2$ decreases sharply, indicating that the photogenerated electrons play a vital role in the photocatalytic ORR. Separately, when EDTA-2Na is added to the system, the equilibrium concentration of $H_2O_2$ is

increased, which is attributed to that the trapping of holes promotes the use of electrons. Noting that little $H_2O_2$ is detected when added BQ to the system, implying that the $\cdot O_2^-$ is an indispensable part during the process of photosynthetic $H_2O_2$. In contrast, the addition of TBA shows almost no influence on $H_2O_2$ production, this is mean that $\cdot OH$ did not participate in the photocatalytic reaction. According to the previous reports[32,33], we conclude that PM-CDs-30 reduces oxygen to produce $H_2O_2$ through a two-electron reaction ($O_2 \rightarrow \cdot O_2^- \rightarrow H_2O_2$), while oxidizes water to releases $O_2$ through a four-electron reaction ($H_2O \rightarrow O_2$).

EPR detection was employed to record hydroxyl radical ($\cdot OH$) and superoxide radical ($\cdot O_2^-$) generation during the photocatalytic reaction process in seawater. As shown in Figure S26, there is no signal of DMPO-$\cdot OH$ observed under darkness and illumination, confirming that no $\cdot OH$ are produced in photocatalytic reaction. That is, the half reaction in which the holes participate, water oxidation reaction, does not produce $H_2O_2$. Notably, six DMPO-$\cdot O_2^-$ signals are observed after light illumination in Figure 5d, suggesting that $\cdot O_2^-$ is the main free radical in the photocatalytic reaction. Based on the above tests and experiments, it can be seen that the generated $H_2O_2$ is obtained by the ORR involving electrons, while the water oxidation process involving holes only releases $O_2$ without any $H_2O_2$ generation.

A typical atmosphere course for $H_2O_2$ production over PM-CDs-30 under visible light irradiation is displayed in Fig. 5e. In $N_2$-saturated seawater obtained by bubbling $N_2$ for 20 min, a sharp decreased in photocatalytic $H_2O_2$ generation is observed. Different from the $N_2$ environment, the production rate of $H_2O_2$ increased in seawater saturated with $O_2$, showing that $O_2$ was involved in the catalytic reaction. This result largely indicates that the photocatalyst PM-CDs-30 catalyzed ORR to synthesise $H_2O_2$ and water oxidation to produce $O_2$.

A series of in situ TPV tests on PM-CDs-30 were performed under different conditions (Fig. 5f). Obviously, the photovoltage intensity of PM-CDs-30 in $O_2$-saturated acetonitrile decreases sharply compared to $N_2$-saturated acetonitrile, but increases greatly when slight water is added. Here, the added $O_2$ interacts with electrons in the ORR, reducing the intensity of the detected signal, while the added water, as a hole sacrificial agent, increasing the amounts of electrons relatively. That is to say, in the photocatalytic reaction, electrons are consumed by oxygen to produce hydrogen peroxide, while holes combined with water to release oxygen. Interesting, when water is substituted by seawater, the photovoltage shows the highest intensity, proving that the salt ions in seawater have a certain promotion effect on PM-CDs-30.

To compare the effects of the main components in seawater on catalytic activity, the synthetic efficiencies of $H_2O_2$ in three separate ionic salts (NaCl, MgCl$_2$, and CaCl$_2$) were plotted in Fig. 5g. For this study, we controlled the concentration of salt ions around their respective concentrations in seawater, according to the international standard of seawater composition[34]. It should be noted that, in the range of seawater concentration, the addition of NaCl, MgCl$_2$, and CaCl$_2$ improve the $H_2O_2$ yield of the photocatalyst.

The electron transfer numbers of PM-CDs-30 during the photocatalytic half reaction were evaluated by the rotating disc electrode (RDE) and rotating-ring disc electrode (RRDE) in seawater. As depicted in Fig. S27, the electron transfer number is calculated by $H_2O_2$ detected in $O_2$-saturated seawater, which is calculated to be 1.82 according to the equation mentioned in the supporting information, suggesting a predominant two-electron pathway for ORR[35,36]. However, the ring current (red line) in Fig. S28 does not change, while the disk current saliently improved when the light was applied to the $N_2$-saturated

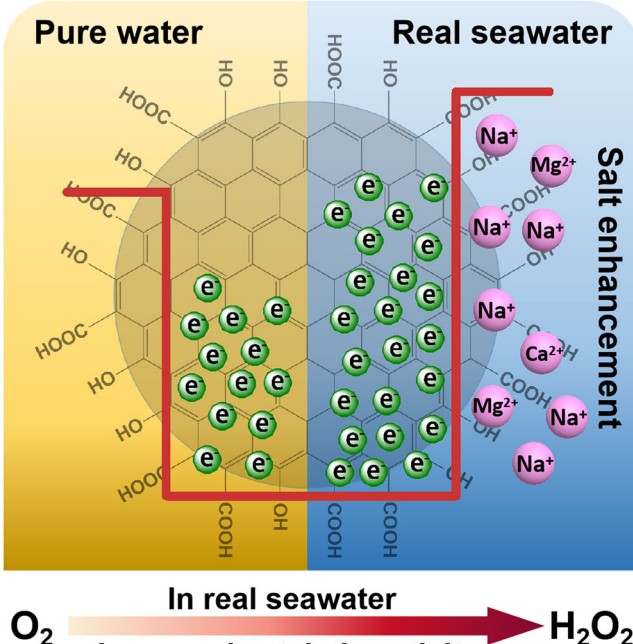

**Fig. 6 Schematic mechanism.** Schematic mechanism of $H_2O_2$ production over PM-CDs-30 through photocatalytic reaction in seawater.

seawater, indicating a four-electron reaction process for water oxygen[37]. Thus, the photocatalytic reaction of PM-CDs-30 in seawater is a two-channel parallel cascade reaction, that is, a parallel combination of two-channel oxygen reduction and four-electron water oxidation reactions.

## Discussion

Based on above results and analysis, the possible mechanism for photocatalysis on PM-CDs-30 is illustrated in Fig. 6. Under the illumination of visible light ($\lambda \geq 420$ nm), PM-CDs-30 can easily absorb visible light to generate electron–hole pairs. The photogenerated electrons are quickly transferred to the CDs, while photogenerated holes remain on the polymer. The electrons on the surface of CDs combine with oxygen in water to form $H_2O_2$, and the holes in the polymer oxidize water to produce $O_2$, which are performed simultaneously as two parallel reactions. As shown in Fig. 6, due to the presence of metal cations in seawater, the functional groups on the surface of CDs are ionized, which enhances the electron sink effect of CDs on electrons. Therefore, CDs can attract and restrain more electrons, which prolongs the separation time of electron and hole, improves the catalytic activity of the photocatalyst for photocatalytic oxygen reduction to produce $H_2O_2$ in real seawater.

In this work, we demonstrate a metal-free composite catalyst PM-CDs-30 with high photocatalytic activity for $H_2O_2$ production in seawater. The superior performance of the photocatalyst originates from the addition of CDs in the catalyst, which increases the time of electron–hole separation. Importantly, the ions in seawater ionize the functional groups on the surface of CDs, which amplifies the electron sink effect of CDs, making the photocatalytic activity in seawater better than in the pure water. PM-CDs-30 photogenerated $H_2O_2$ at a rate of 1776 µmol g$^{-1}$ h$^{-1}$ in real seawater, which is 4.8 times than that of the pure polymer (PM-CDs-0). The SCC efficiency of PM-CDs-30 can reach 0.21%, which is an important breakthrough in the field of seawater photocatalysis synthesis of hydrogen peroxide. Moreover, theoretical calculations are consistent with the experiments, which

give the optimum reaction conditions and the maximum reaction rate under these conditions. Our catalytic system not only provides an idea for photocatalytic $H_2O_2$ production in seawater, but also provides a promising way for the design, selection and optimization of photocatalysts.

## Methods

**Materials**. All chemicals were not extra purified before use. $Na_2CO_3$ (AR, 98%) and $HNO_3$ (AR, 98%) were purchased from Aladdin. Procyanidins (PC, AR, 98%) was purchased from Yuanye. 4-Methoxybenzaldehyde (MB, AR, 98%) was purchased from Sinopharm Group chemical Reagent Co., Ltd. The seawater was taken from the Yellow Sea.

**Synthesis of CDs**. The CDs were fabricated by an electrochemical method[20,25]. Briefly, two graphite rods were cleaned in deionized water by ultrasonic agitation for 30 min for three times. Then, the cleaned graphite rods were fixed as cathode and anode, respectively, in the ultrapure water. Next, 30 V was applied across the electrodes using a direct current (DC) power supply. The electrolytic process was running for the 20 days under strongly stirring. After that, the black solution was filtered with a four-stage membrane separation and filtration system (DMJ60) to obtain CDs solution with the particle diameter of less than 50 nm. Finally, the filtered CDs solution was freeze-dried to obtain CDs powder.

**Synthesis of PM-CDs-$x$ catalysts**. The organic polymer composite systems PM-CDs-$x$ were synthesized according to the literature by a combination of sol–gel, crosslinking polymerization and freeze-drying technology[33]. First of all, 0, 10, 30, 60, and 100 mg of CDs powders were respectively dissolved in 10 mL water to obtain the CDs solutions of different concentrations. Then, 0.5 g of PC and 0.1 g of $Na_2CO_3$ were dissolved in 10 mL CDs solutions, and then 0.9 g of MB was added. After that, the mixture was sealed and transferred to an oil bath at 80 °C for 48 h to form a gelatinous precursor. The precursor was washed with ethanol and water to remove residual organic solvents. Finally, after freezing with liquid nitrogen freezing (−196 °C, 10 min) and freeze-drying in a vacuum chamber (−53 °C, 0.113 mbar), the crosslinked organic semiconductor photocatalyst PM-CDs-$x$ were obtained ($x$ is the weight of the added CDs), which were successively labeled as PM-CDs-0, PM-CDs-10, PM-CDs-30, PM-CDs-60, and PM-CDs-100.

**Experiment of photocatalysis**. The photocatalytic activity of the photocatalysts was assessed by a multichannel photochemical reaction system (PCX-50B, Beijing Perfectlight Co., Ltd, China) equipped with a visible light source ($\lambda \geq 420$ nm), without the addition of sacrificial agents and other cocatalysts. In a typical reaction, 10 mg of the catalyst was dispersed in a quartz bottle (50 mL) containing 20 mL of seawater and then it was exposed to visible light for 5 h without seal in air at room temperature. The rotation speed was 180 r.p.m. The optimal activity and reaction mechanism of the catalyst were explored by changing the light condition (with and without light), the illumination time (5, 12, 24, 48, 72, 84, and 96 h), reaction environment (water or seawater), the excitation wavelength ($\lambda = 365$, 420, 535, and 620 nm), and the sacrificial agent (AgNO₃, EDTA-2Na, TBA, and BQ) (2 mM). After each reaction, the catalyst was washed and dried for stability testing (5 h for each cycle).

Experiments on the influence of rotating speed on catalytic activity were carried out at different rotating speeds, which were 40, 80, 120, 160, and 200 r.p.m., respectively. Here, 10 mg catalyst was dispersed in 20 mL seawater (or pure water), and the quartz bottle (50 mL) was not sealed. The catalytic reaction took places in air at room temperature under visible light irradiation ($\lambda \geq 420$ nm) for 5 h. For the experiment of the relationship between oxygen partial pressure and catalytic activity, the catalytic system was sealed, and then oxygen (0, 4, 8, 12, and 16 mL) was injected into it to control the oxygen partial pressure of the catalytic system. Then, the catalytic reaction was carried out at 180 r.p.m. under visible light irradiation ($\lambda \geq 420$ nm) for 5 h.

For the reaction atmosphere, $N_2$ (or $O_2$) was used to bubble in seawater for 20 min, and then the catalytic activity of the catalyst was tested under different atmospheres under airtight condition. As for the effect of different types of salt on the activity, different concentrations of NaCl solution, $MgCl_2$ solution, and $CaCl_2$ solution were employed (the concentrations were within the range of seawater concentration). Here, difference significance test of statistics method was adopted in the final analysis. The experiments were analyzed using one-way analysis of variance (ANOVA) followed by Tukey–Kramer post hoc test, and the analytical and experimental replicates were used three time at least.

In order to detect the change of hydrogen peroxide yield with temperature, 20 mL seawater containing 10 mg PM-CDs-30 was performed at different temperatures (20, 40, 60, and 80 °C) under visible light irradiation for 5 h. In addition, the temperature change of the reaction solution (20 mL seawater with 10 mg catalyst) during the photocatalytic reaction was monitored by an infrared camera under room temperature and unsealed conditions.

To monitor the evolution of $O_2$, 10 mg of the catalyst was dispersed in a quartz bottle (50 mL) containing 20 mL of water and then 10 mM of AgCl was added. The system was bubbled by $N_2$ for 30 min. The sealed reaction system reacted 2, 4, and 6 h under light irradiation, and then gas chromatography was used to detect the change of oxygen.

At the end of the experiment, the solution was centrifuged, filtered, and then the yield of $H_2O_2$ was determined by 0.1 M $KMnO_4$.

## Data availability

The experimental data that support the findings of this study are available from the corresponding author upon reasonable request. Source data are provided with this paper.

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

## Acknowledgements

This work is supported by National MCF Energy R&D Program (2018YFE0306105), National Key Research and Development Project of China (2020YFA0406104 and 2017YFA0204800), Innovative Research Group Project of the National Natural Science Foundation of China (51821002), National Natural Science Foundation of China (51725204, 21771132, 51972216, 52041202), Natural Science Foundation of Jiangsu Province (BK20190041), Key-Area Research and Development Program of GuangDong Province (2019B010933001), Collaborative Innovation Center of Suzhou Nano Science & Technology, the Priority Academic Program Development of Jiangsu Higher Education Institutions (PAPD), and the 111 Project. Finally, we would like to thank Prof. Haiping Lin for his contribution to the calculation part of this work.

## Author contributions

Q.W., J.C., and Z.K. designed and planned the experiments. Q.W., X.W., and Y.L. prepared the samples. Y.Z, H.W., H.H., and L.F. carried out the photochemical experiment. Q.W., J.C., H.H, Y.L., M.S., and Z.K analyzed the data. All the authors participated in the interpretation and discussion of the results. Q.W., Y.L., M.S., and Z.K. wrote the article. All co-authors contributed to the manuscript.

## Competing interests

There are no conflicts of interest to declare.
