## [Peer Review File · Nature Communications]

REVIEWER COMMENTS

Reviewer #1 (Remarks to the Author):

Recommendation: Accept after minor revisions.

In the manuscript, the authors proposed a theoretical model of electron sink in CDs. The in-situ transient photovoltage test was used to demonstrate the salt-protective electron sink effect, which is a vital and emerging tool for mechanism study. By combining the CDs with organic polymers, the authors prepared metal-free catalyst PM-CDs-30 to achieve highly efficiency production of hydrogen peroxide in seawater, even larger than in freshwater, which is a breakthrough advance in photocatalysis in salt ion solutions. This study is interesting. The manuscript is prepared with high quality. Therefore, I recommend that the manuscript be accepted with minor modifications.

1. In general, the highest occupied molecular orbital (HOMO) and lowest unoccupied molecular orbital (LUMO) of most organics are identified by cyclic voltammetry method. The HOMO and LUMO of PM-CDs-30 should be mentioned to verify the energy level of the catalyst.

2. The schematic in Figure 1c may not match the description. Please correct it or provide more detailed explanation.

3. In this paper, the authors stated that "CDs can perform electrocatalytic oxygen reduction reaction by two-electron channel", and provided the electrochemical properties of oxygen reduction reaction for CDs. However, did the CDs have light-enhanced activity in oxygen reduction reaction? Please verify it.

4. Specific surface area is an important data and the specific surface area of all catalysts need be provided. Whether the changes of the specific surface area significant affect the catalytic properties?

5. In Figure S18c, the electron sink effect on the catalyst becomes more obvious with the increase of the amount of CDs; however, in Figure 3b, with the increase of the amount of CDs, the enhancing effect of the catalyst on the properties in seawater becomes weaker. Why?

Reviewer #2 (Remarks to the Author):

The paper of Kang and co-workers describes the use of a polymer composite containing CDs for the hydrogen peroxide photoproduction in seawater. The authors have used an appropriate set of characterization techniques and all the experiments have been well performed. The reported QY of 0.54% do not outperform already reported works (for instance, Nat. Mater. 18, 985–993 (2019)), but it is interesting that this system works in seawater (and do so even better as compared to pure water). This is a nice and interesting work, useful and important not only for the community interested in CDs but for all the researchers working in the catalysis area. I can recommend publication in Nature Communication if the authors consider and address the following minor points:

The references cited are overall adequate, but it would be important for the reader that the authors include a brief discussion of the performance's indicator in the context of the state of the art. The discussion should be extended to the solar-to-chemical conversion efficiency of their system, which is currently not present in the paper and should be included in the manuscript. In this way, the authors can better clarify and highlight the strength of their work, and they should also expand the comments in the outlook/future work section.

The authors report that the best system is the PM-CDs-30 and that, after 72 hours of reaction, the yield of hydrogen peroxide showed a slightly downward trend:

(i) The authors suggest that the decomposition of hydrogen peroxide may be the reason for such downward trend, which I believe that it is a reasonable explanation. The same authors have previously reported that the decomposition of hydrogen peroxide assists the water splitting reaction when using CDs. Have the author tried to decompose hydrogen peroxide with this system? I think that they should include this experiment in the SI. Also, the authors should provide proof that across the reaction time there is no leaching of the CDs.

(ii) Increasing the content of CDs results in the increase of the absorption in NIR region, but a content higher of 30 is detrimental for the catalysis. Do the authors have an explanation for this? Have they observed, for instance, CD agglomeration at higher loading?

The authors performed the experiments at room temperature, but have they observed a dependence of the photocatalytic activity with the photoreaction temperature?

It would be helpful for the reader if the authors will include a schematic figure detailing the composition/structure of the PM-CDs in the first part of the paper.

Reviewer #3 (Remarks to the Author):

This paper reports a study on the photocatalytic H₂O₂ generation by carbon dots (CDs)-doped polymeric photocatalyst in seawater. H₂O₂ is an emerging fuel and environmentally friendly oxidant that is currently used in many fields, but the current production processes of H₂O₂ are energy and cost consumptive and generate a lot of wastes. Therefore, sustainable processes to manufacture H₂O₂ are needed. The major finding of this work is that the PM-CD photocatalyst shows a H₂O₂ yield at 1776 $\mu\text{mol/g/h}$ in seawater, higher than that (1246 $\mu\text{mol/g/h}$) in water. The authors suggest that the salt ions (i.e., Na⁺, Ca²⁺) in the seawater enhance the electron-withdrawing property of the functional group of CDs that increases electron extraction rate under excitation (line 102-103). However, the addition of Na⁺, Ca²⁺, or Mg²⁺ does not significantly increase the H₂O₂ photoproduction rates (Figure 4g), although the authors still claim a large increase of H₂O₂ upon their addition (line 372). Results in Figure 4g contradicts with suggested role of Na⁺, Ca²⁺, Mg²⁺ in seawater as promoters for electron extraction in photocatalysis involving CDs. The current mechanistic discussion behind the seawater effect is very speculative (line 98-118, and line 317-322) and should be improved by theoretical calculation and considerations. Additionally, it is known that high salt concentrations/ionic strength can destabilize nanocatalysts by aggregation as a result of electric double layer suppression. There is no such consideration and the enhanced H₂O₂ formation in seawater appears to disagree with catalyst aggregation in seawater. There are also important considerations in the revision of the manuscript.

1. The performance of photocatalyst in seawater should be compared to those in existing work using seawater, to explicitly highlight uniqueness of this work. The AQYs can also be compared among related photocatalytic H₂O₂ generation studies.

2. It is suggested that H₂O acts the electron donor and forms O₂ (i.e., organic donor free). The authors should consider to measure O₂ evolution to confirm this important claim.

3. The photocatalyst PM-CDs-30 actually shows some oxidation after photocatalysis as in Figure S17b. The the relative C-O group fraction increases as compared to that in the parent photocatalyst (Figure 2b). Could this oxidation provide the electron for O₂ reduction and H₂O₂ formation? The measurement of O₂ formation (H₂O as the electron donor) should be quantitatively compared to that of H₂O₂ formation, so that other electron sources can be evaluated and establish H₂O as the electron donor in the system.

4. Line 242 the suggested H₂O₂ decomposition should be evaluated/measured in the presence of photocatalysts with added H₂O₂.

5. Line 255 considering the amount of H₂O₂ formed >1000 μmole (Figure 3f), the required O₂ concentration would exceed 50 mM (20 mL water) in water. How could this be possible given the solubility limit of O₂? Did the experiment actively bubble air/O₂ in the solution? Please clarify.

6. CD is suggested to act as the electron trap. To confirm this, the photoluminescence indicative of electron-hole re-combination should be measured for the various photocatalysts made.

7. The figure labels (including x, y axis labels) especially Figure 3&4 are too small to read. The sub-figures in Figure 3 are mis-labeled in the caption and Figure 3b&c are not introduced in the caption. Typos in the Figure 3 labels should be corrected. In Figure 3d&e, the model calculation should be expressed as lines and measurement data in points. Additionally, more description of the experimental conditions used in Figure 3&4 is useful, as the photocatalysis experiments described in method section is very generally, for example, not covering how rotation and oxygen partial pressure were conducted.

8. There are many errors (grammars, typos, mistakes) in current text that should be carefully corrected.

Responses to the Referees' Comments

In the first place, we sincerely thank the editors and referees' work on our manuscript. We appreciate your recognition of our work. Herein, we respond to the referees' insightful comments and suggestions in detail.

Referee: #1

Comments to the Author

In the manuscript, the authors proposed a theoretical model of electron sink in CDs. The *in-situ* transient photovoltage test was used to demonstrate the salt-protective electron sink effect, which is a vital and emerging tool for mechanism study. By combining the CDs with organic polymers, the authors prepared metal-free catalyst PM-CDs-30 to achieve highly efficiency production of hydrogen peroxide in seawater, even larger than in freshwater, which is a breakthrough advance in photocatalysis in salt ion solutions. This study is interesting. The manuscript is prepared with high quality. Therefore, I recommend that the manuscript be accepted with minor modifications.

Comment 1:

In general, the highest occupied molecular orbital (HOMO) and lowest unoccupied molecular orbital (LUMO) of most organics are identified by cyclic voltammetry method. The HOMO and LUMO of PM-CDs-30 should be mentioned to verify the energy level of the catalyst.

Response 1:

Thanks for your valuable comment.

For present composite catalysts, we used cyclic voltammetry (CV) method to calculate the HOMO (or conduction band E_c) and LUMO (or valence band E_v) and verify the location of the conduction band and the valence band of pure polymer and the composite catalyst. The CV curves of pure polymer PM-CDs-0 and the composite catalyst PM-CDs-30 are shown in Figure S13. As shown in the Figure S13, the pure

polymer and the composite catalyst have the similar energy band levels. Here, for PM-CDs-30, the valance band energy was calculated to be -6.01 eV, while the value of conduction band energy level was calculated to be -4.03 eV.

We have revised our paper, which are shown as follows:

1.3 Cyclic voltammetry (CV) measurement

The cyclic voltammetry (CV) method was carried by a standard three-electrode system with CHI 760E workstation. The sample-modified (PM-CDs-0 and PM-CDs-30) glassy carbon (GC), Ag/AgCl electrode and carbon electrode were used as the work electrode, reference electrode and counter electrode, respectively. In this experiment, 4 μL catalyst solution (2 mg mL^{-1}) and 5 μL Nafion solution (0.5 wt%) were dropped onto the working area of a cleaned GC electrode and put naturally to dry. The CV curves were measured in N_2 -saturated 0.1 M BMIMPF6 solution with a scan rate of 50 mV s^{-1} . Ferrocene was added into the above solution as an internal standard with a concentration of 1 mg mL^{-1} . The energy levels of the catalyst were calculated from the onset oxidation ($E_{\text{onset}}^{\text{ox}}$), reduction ($E_{\text{onset}}^{\text{red}}$) potential and the onset oxidation potential of ferrocene ($E_{\text{ferrocene}}$).

Figure S13. (a) CV curve of PM-CDs-0-modified glassy carbon (GC) electrode in N_2 -saturated anhydrous acetonitrile (0.1 M BMIMPF6) with ferrocene as the internal standard. (b) CV curve of PM-CDs-30-modified GC electrode in N_2 -saturated anhydrous acetonitrile (0.1 M BMIMPF6) with ferrocene as the internal standard.

Comment 2:

The schematic in Figure 1c may not match the description. Please correct it or

provide more detailed explanation.

Response 2:

Thanks for your precise viewpoint. We have corrected Figure 1c according to the description in the article. When the metal salt (M^+) was added, the carboxyl group can be ionized, and the electronegativity (δ_1^-) of oxygen in carbon group increase, which increase the surface charges and electron sink barrier of CDs. Therefore, we modified the amounts of electrons in the electron sink after ionized in Figure 1c.

The revised Figure is shown as follows.

Figure 1. Structure and property of CDs. (a) TEM image of CDs with the particle size distribution inserted. (b) HRTEM image of CDs with FT-IR spectrum inserted. (c) Schematic diagram of the electron sink model for CDs. (d) TPV curves of CDs powders before and after adding NaCl.

Comment 3:

In this paper, the authors stated that “CDs can perform electrocatalytic oxygen reduction reaction by two-electron channel”, and provided the electrochemical properties of oxygen reduction reaction for CDs. However, did the CDs have light-enhanced activity in oxygen reduction reaction? Please verify it.

Response 3:

Thank you for your comments.

The influence of light on the oxygen reduction activity of CDs was detected and the result is shown in Figure S6b. As we can see from the Figure S6b, the oxygen reduction activity of CDs was not significantly changed by light, showing that CDs have no light-enhanced activity in oxygen reduction reaction.

The revised parts are shown as follows.

The TPV tests of CDs mixed with other metal salt ions shown in Figure S5 display the same phenomenon as that added with NaCl. Another important property in Figure S6 reveals that CDs can perform electrocatalytic oxygen reduction reaction by two-electron channel with no photoelectric enhancement effect, suggesting that it can be used as the oxygen reduction reaction site in catalytic system. Thus, the functional groups (such as -OH, C=O, -COOH) on the surface of CDs not merely play an extraordinary role in the design and performance regulation of catalysts, but also effectively trap electrons through its electron sink effect in seawater to accelerate the electron transfer and prevent electron-hole recombination, thereby improving the photocatalytic activity of the catalyst in seawater.

Figure S6. (a) Linear sweep voltammetry (LSV) curve of CDs-loaded electrode toward oxygen reduction reaction (ORR). (b) LSV curves of CDs-loaded electrode toward ORR with and without light.

Comment 4:

Specific surface area is an important data and the specific surface area of all catalysts need be provided. Whether the changes of the specific surface area significant affect the catalytic properties?

Response 4:

Thank you for your precise suggestion.

The specific surface areas of all catalysts are provided in Table S2. As can be seen from the Table S2, the addition of CDs in the pure polymer increases the specific surface area of the polymer composite catalyst. However, the specific surface area of the composite catalysts with different amounts of CDs is similar, about $5\text{m}^2\text{g}^{-1}$. So, in present composite catalyst system, the specific surface area has no significant effect on the activity of the catalysts.

The revised part is shown as follows:

In addition, a series of basic characterizations, such as the size distribution (Figure S8), XRD patterns (Figure S9a), FT-IR spectra (Figure S9b), XPS spectra (Figure S10), elemental analysis (Table S1) and specific surface area (Table S2), declared that PM-CDs-30 was an amorphous carbon structure composed of C, H, O elements with abundant functional groups.

Table S2. The specific surface area of PM-CDs-x samples.

Photocatalyst	Specific surface area
	$[\text{m}^2\text{g}^{-1}]$
PM-CDs-0	1.5076
PM-CDs-10	4.9504
PM-CDs-30	4.9129
PM-CDs-60	5.1849

Comment 5:

In Figure S18c, the electron sink effect on the catalyst becomes more obvious with the increase of the amount of CDs; however, in Figure 3b, with the increase of the amount of CDs, the enhancing effect of the catalyst on the properties in seawater becomes weaker. Why?

Response 5:

Thank you for your valuable advice.

In the paper, the increase of CDs content enhances the electron sink effect of the catalyst, and the electron sink effect of the catalyst was enhanced. Yet, the catalytic activity does not increase. This is because the improvement of catalytic activity includes not only the electron sink effect of CDs, but also the electron transfer rate, photo-response performance and charge extraction capacity of the catalyst. As described in the manuscript, photocatalyst PM-CDs-30 has the lowest impedance, the best photoelectric response and the highest charge excitation. Therefore PM-CDs-30 exhibits the best catalytic activity.

Referee: #2

Comments to the Author

The paper of Kang and co-workers describes the use of a polymer composite containing CDs for the hydrogen peroxide photoproduction in seawater. The authors have used an appropriate set of characterization techniques and all the experiments have been well performed. The reported QY of 0.54% do not outperform already reported works (for instance, Nat. Mater. 18, 985–993 (2019)), but it is interesting that this system works in seawater (and do so even better as compared to pure water). This is a nice and interesting work, useful and important not only for the community interested in CDs but for all the researchers working in the catalysis area. I can recommend publication in Nature Communication if the authors consider and address the following minor points:

Comment 1:

The references cited are overall adequate, but it would be important for the reader that the authors include a brief discussion of the performance's indicator in the context of the state of the art. The discussion should be extended to the solar-to-chemical conversion efficiency of their system, which is currently not present in the paper and should be included in the manuscript. In this way, the authors can better clarify and highlight the strength of their work, and they should also expand the comments in the outlook/future work section.

Response 1:

Thanks for your precise comments.

The solar-to-chemical conversion efficiency of the photocatalytic system has been calculated and discussed in the revised manuscript. Furthermore, due to only a few studies on the photocatalytic production of hydrogen peroxide in seawater, the comparison of the catalytic activities of different photocatalysts systems in water and seawater had been provided in Table S4.

The revised parts are shown as follows:

It turned out that this ideal composite catalyst exhibits the expected photocatalytic

capacity to generate H₂O₂ in seawater, and the catalytic activity of the optimal composite photocatalyst is approximately 4.8 times higher than the pure polymer photocatalyst in seawater. In particular, the maximum yield of H₂O₂ for the optimal catalyst PM-CDs-30 was 1776 μmol g⁻¹h⁻¹ in seawater, the apparent quantum yield (AQY) was as high as 0.54% at 630 nm in seawater, and the solar-to-chemical conversion (SCC) efficiency could reach 0.21% in seawater.

Relatively speaking, PM-CDs-30 shows the best activity in both pure water and seawater solution among the catalysts studied in this work, and its photocatalytic activity in seawater (1776 μmol g⁻¹h⁻¹) was much higher than that in pure water (1340 μmol g⁻¹h⁻¹). And the solar-to-chemical conversion (SCC) efficiency of PM-CDs-30 could reach 0.21% in seawater. To further explore the photocatalytic activity of the catalysts in real seawater and pure water, the difference in hydrogen peroxide production rate between seawater and pure water is displayed in Figure 3b.

Conclusion

In this work, we demonstrated a metal-free composite catalyst PM-CDs-30 with high photocatalytic activity for H₂O₂ production in seawater. The superior performance of the photocatalyst originates from the addition of CDs in the catalyst, which increases the time of electron-hole separation. Importantly, the ions in seawater ionize the functional groups on the surface of CDs, which amplifies the electron sink effect of CDs, making the photocatalytic activity in seawater better than in the pure water. PM-CDs-30 photo-generated H₂O₂ at a rate of 1776 μmol g⁻¹h⁻¹ in real seawater, which is 4.8 times higher than that of the pure polymer (PM-CDs-0). The SCC efficiency of PM-CDs-30 could reach 0.21%, which is an important breakthrough in the field of seawater photocatalysis synthesis of hydrogen peroxide. Moreover, theoretical calculations are consistent with the experiments, which gives the optimum reaction conditions and the maximum reaction rate under these conditions. Our catalytic system not only provides a new idea for photocatalytic H₂O₂ production in seawater, but also provides a promising way for the design, selection and optimization of photocatalysts.

1.5 Solar-to-chemical conversion (SCC) efficiency

In the experiment, the multichannel photochemical reaction system equipped with the visible light ($\lambda \geq 420$ nm) as the light source and PM-CDs-30 (10 mg) as the catalyst was used to calculate the solar-to-chemical conversion (SCC) efficiency. After 5 h of illumination, the total incident power over the 8.04 cm² irradiation area was 34.8 mW cm⁻². So that the total input energy in 5 h was:

$$E_{\text{soalr}}=5036.2 \text{ J}$$

During the photocatalytic reaction, 88.8 μmol H₂O₂ was detected, which indicated that the energy of produced hydrogen peroxide was:

$$E_{\text{H}_2\text{O}_2} = n(\text{H}_2\text{O}_2) \times \Delta G(\text{H}_2\text{O}_2) = 88.8 \times 10^{-6} \times 117 \times 10^3 \text{ J} = 10.4 \text{ J}$$

The SCC conversion efficiency of PM-CDs-30 was determined to be:

$$\text{SCC} = \frac{E_{\text{H}_2\text{O}_2}}{E_{\text{solar}}} \times 100\% = \frac{10.4 \text{ J}}{5036.2 \text{ J}} \times 100\% = 0.21\%$$

Table S4. Comparison of the catalytic activities of different photocatalyst systems in the literatures.

Photocatalyst	Condition	H ₂ O ₂ μmol $\text{h}^{-1}\text{g}^{-1}$	AQY at 420 nm	SCC %	Ref.
PM-CDs-30	Real seawater	1776	0.99	0.21	This work
m-WO ₃ /FTO		2			
photoanode-Co ^{II} (Ch)/CP cathode	Artificial seawater	mmol L ⁻¹ h ⁻¹	-	0.55	5
TiO ₂	4% NaCl solution	90	-	-	6
FeO(OH)/BiVO ₄ /FTO		6.5			
photoanode-Co ^{II} (Ch)/carb on paper cathode	Artificial seawater	mmol L ⁻¹ h ⁻¹	-	0.89	7
Au/BiVO ₄	O ₂ -saturated water	2.412	0.24%	-	8
g-C ₃ N ₄ /NaBH ₄	Water	170	4.3%	0.26	9

g-C ₃ N ₄ /BDI	O ₂ -saturated water	17	2.6%	0.13	10
g-C ₃ N ₄ /PDI/rGO	O ₂ -saturated water	23.4	6.1%	0.2	11
C ₃ N ₄ / anthraquinone	O ₂ -saturated water	361	4.8%	0.178	3
RF523	O ₂ -saturated water	82.5	6%	0.5	12

Supplemental References

5. Mase, K., Yoneda, M., Yamada, Y. & Fukuzumi, S. Seawater usable for production and consumption of hydrogen peroxide as a solar fuel. *Nat. Commun.* **7**, 11470 (2016).
6. Harada, H. Isolation of hydrogen from water and/or artificial seawater by sonophotocatalysis using alternating irradiation method. *Int. J. Hydrogen Energ.* **26**, 303-307 (2001).
7. Mase, K., Yoneda, M., Yamada, Y. & Fukuzumi, S. Efficient photocatalytic production of hydrogen peroxide from water and dioxygen with bismuth vanadate and a cobalt(II) chlorin complex. *ACS Energy Lett.* **1**, 913–919 (2016).
8. Hirakawa, H. *et al.* Au nanoparticles supported on BiVO₄: effective inorganic photocatalysts for H₂O₂ production from water and O₂ under visible light. *ACS Catal.* **6**, 4976–4982 (2016).
9. Zhu, Z., Pan, H., Murugananthan, M., Gong, J. & Zhang, Y. Visible light-driven photocatalytically active g-C₃N₄ material for enhanced generation of H₂O₂. *Appl. Catal. B-Environ.* **232**, 19–25 (2018).
10. Kofuji, Y. *et al.* Graphitic carbon nitride doped with biphenyl diimide: efficient photocatalyst for hydrogen peroxide production from water and molecular oxygen by sunlight. *ACS Catal.* **6**, 7021–7029 (2016).
11. Kofuji, Y. *et al.* Carbon nitride–aromatic diimide–graphene nanohybrids: metal-free photocatalysts for solar-to-hydrogen peroxide energy conversion with 0.2% Efficiency. *J. Am. Chem. Soc.* **138**, 10019–10025 (2016).
12. Shiraishi, Y. *et al.* Resorcinol–formaldehyde resins as metal-free semiconductor photocatalysts for solar-to-hydrogen peroxide energy conversion. *Nat. Mater.* **18**, 985–993 (2019).

Comment 2:

The authors report that the best system is the PM-CDs-30 and that, after 72 hours of reaction, the yield of hydrogen peroxide showed a slightly downward trend:

(i) The authors suggest that the decomposition of hydrogen peroxide may be the

reason for such downward trend, which I believe that it is a reasonable explanation. The same authors have previously reported that the decomposition of hydrogen peroxide assists the water splitting reaction when using CDs. Have the author tried to decompose hydrogen peroxide with this system? I think that they should include this experiment in the SI. Also, the authors should provide proof that across the reaction time there is no leaching of the CDs.

(ii) Increasing the content of CDs results in the increase of the absorption in NIR region, but a content higher of 30 is detrimental for the catalysis. Do the authors have an explanation for this? Have they observed, for instance, CD agglomeration at higher loading?

Response 2:

Thanks for your valuable advice.

(i) The experiment for the decomposition of hydrogen peroxide experiments has been carried out and displayed in Figure S18. As seen from Figure S18, the catalyst PM-CDs-30 has slight decomposition ability of hydrogen peroxide.

In order to detect whether CDs leach after the catalytic reaction, a 100 nm filter was used to filter the reaction solution (before and after photocatalytic reaction) to obtain the filter liquor. Then, the presence of CDs in the filter liquor was examined to determine whether CDs leached during the catalytic reaction. CDs were not found in the filter liquor by a lot of TEM tests. In addition, the carbon content in the filter liquor of different samples was lower than the detection limit before and after photocatalytic reaction (Table R1). Based on all above results, it can be said that across the reaction time there is no leaching of the CDs.

(ii) First of all, as shown in Figure R1, the TEM images of PM-CDs-60 and PM-CDs-100 show that with the increase in the amounts of CDs added, the CDs disperse more density in the polymer, but there is no obvious aggregation phenomenon at higher CDs loading. Besides, the light absorption of catalyst is one of the factors affecting its catalytic activity, but it is not the only important factor. Although the increase CDs content results in the increase of the absorption in NIR region, it does not improve other properties of the catalyst at a high loading of CDs,

such as the impedance, light response performance and charge extraction, which are important factors affecting the photocatalytic performance of the catalyst. We can see from the experiments and test results that PM-CDs-30 has the minimum impedance, the best photo-response performance and the maximum charge excitation. These factors jointly determined that PM-CDs-30 exhibits the best catalytic activity.

The revised parts are shown as follows:

1.6 Electrochemical measurement of the PM-CDs-30 for decomposition of H₂O₂

The H₂O₂ decomposition behavior of PM-CDs-30 was measured by cycle voltammetry (CV) in 0.2 M (pH = 7) phosphate buffered solution with 25 mM H₂O₂. A standard three-electrode system with CHI 760E workstation was used. The carbon electrode and the saturated calomel electrode were used as the counter electrode and the reference electrode, respectively. The bare glassy carbon (GC) electrode (3 mm diameter) and PM-CDs-30-modified GC were used as the work electrodes, which are used as the control group and experiment group respectively. Here, 4 μL catalyst solution (2 mg mL⁻¹) and 5 μL of 0.5 wt % Nafion solution were dropped onto the working area of a cleaned GC electrode and put naturally to dry. The CV curves were measured under darkness with a scan rate of 50 mV s⁻¹ and the results are shown in Figure S18a.

1.7 Degradation of H₂O₂ by PM-CDs-30

10 mg PM-CDs-30 was dispersed in 20 mL water with 50 μmol H₂O₂ added. The reaction vessels were then placed under dark conditions and stirred. The content of H₂O₂ in the solution was detected after 0, 6, 12, and 24h. The change of H₂O₂ content is shown in Figure S18b.

Figure S18. (a) Cyclic voltammograms curves of bare GC and PM-CDs-30 modified GC electrodes in 0.2 M (pH=7) phosphate buffered 25 mM H₂O₂ solution at a scan rate of 50 mV s⁻¹ (without light). (b) Change of H₂O₂ content in solution over time under dark condition.

Table R1. The amount of element C in filter liquor of different samples before and after photocatalytic reaction.

Sample	C (%)	
	Before	After
PM-CDs-0	BDL	BDL
PM-CDs-10	BDL	BDL
PM-CDs-30	BDL	BDL
PM-CDs-60	BDL	BDL
PM-CDs-100	BDL	BDL

BDL: below detected limit.

Figure R1. (a) TEM image of PM-CDs-60. (b) TEM image of PM-CDs-100.

Comment 3:

The authors performed the experiments at room temperature, but have they observed a dependence of the photocatalytic activity with the photoreaction temperature?

Response 3:

Thank you for your precise suggestions.

The experiments on changes in catalytic activity and temperature have been done and conducted in Figure S17. As can be seen from Figure S17a, the catalytic activity increases slightly with the increase of temperature below 40 °C. However, at higher temperature, the yield of hydrogen peroxide decreases greatly due to the decomposition of hydrogen peroxide and other factors. Thus, at relatively low temperatures, the increase of reaction temperature has a slight promoting effect on the reaction activity.

In the other hand, our photocatalytic experiments were carried out in a multi-channel photocatalytic reactor at constant room temperature. The Infrared camera was used to monitor the temperature change of the reaction solution during the reaction at room temperature (Figure S17b, c, d), it was found that the temperature of the catalytic system did not change significantly during the photocatalytic reaction at room temperature. Combined with the effect of temperature on catalytic activity, it can be seen that the effect of reaction temperature on catalytic activity is very small when the photocatalytic reaction is carried out at room temperature. Thus, the effect of the photoreaction temperature on the photocatalytic activity might be excluded.

The revised parts are shown as follows:

Figure S17. (a) Temperature course of H₂O₂ photoproduction by PM-CDs-30. (b) Infrared temperature maps of the reaction system before catalytic reaction at room temperature. (c) Infrared temperature maps of the reaction system after catalytic reaction at room temperature for 2 h. (d) Infrared temperature maps of the reaction system after catalytic reaction at room temperature for 5 h.

Comment 4:

It would be helpful for the reader if the authors will include a schematic Figure detailing the composition/structure of the PM-CDs in the first part of the paper.

Response 4:

Thank you for your precise viewpoint.

The schematic of the composition for PM-CDs was added in the first part of the paper (Schematic 1). The reviewed parts are shown as follows:

Schematic 1. The synthesis process of organic polymer composite systems.

As a new star in photocatalytic field, carbon dots (CDs) have been proved to be a

co-catalytic active site and/or a good electron acceptor/donor material, and show unique an ability to improve the catalytic efficiency in the photocatalytic system²⁰⁻²³. Herein, as shown in Schematic 1, we reported a phenolic condensation approach, in which CDs, organic dye molecule procyanidins and 4-methoxybenzaldehyde were composed into metal-free photocatalyst (PM-CDs-x) for the photocatalytic production of H₂O₂ in seawater. Notably, we proposed and proved the electron sink effect of CDs, and this electron sink effect would grow with the addition of metal cations. So, the catalyst could extract more electrons under light excitation, effectively hindering the electron-hole recombination.

Referee: #3

Comments to the Author

This paper reports a study on the photocatalytic H₂O₂ generation by carbon dots (CDs)-doped polymeric photocatalyst in seawater. H₂O₂ is an emerging fuel and environmentally friendly oxidant that is currently used in many fields, but the current production processes of H₂O₂ are energy and cost consumptive and generate a lot of wastes. Therefore, sustainable processes to manufacture H₂O₂ are needed. The major finding of this work is that the PM-CD photocatalyst shows a H₂O₂ yield at 1776 $\mu\text{mol/g/h}$ in seawater, higher than that (1340 $\mu\text{mol/g/h}$) in water.

Comment 1:

The authors suggest that the salt ions (i.e., Na⁺, Ca²⁺) in the seawater enhance the electron-withdrawing property of the functional group of CDs that increases electron extraction rate under excitation (line 102-103). However, the addition of Na⁺, Ca²⁺, or Mg²⁺ does not significantly increase the H₂O₂ photoproduction rates (Figure 4g), although the authors still claim a large increase of H₂O₂ upon their addition (line 372). Results

Response 1:

Thank you for your precise viewpoint.

Figure 4g shows that the maximum hydrogen peroxide generation rate can be increased to 1708 $\mu\text{mol g}^{-1}\text{h}^{-1}$ after adding Mg²⁺, which is 1.27 times of the initial rate (1340 $\mu\text{mol g}^{-1}\text{h}^{-1}$). This is a breakthrough for the production of hydrogen peroxide in seawater. So, we concluded the addition of Na⁺, Ca²⁺, or Mg²⁺ increase the H₂O₂ photoproduction rates. Furthermore, we have revised the description of the production rate, and the revised parts are shown as follows:

To compare the effects of main components in seawater on catalytic activity, the synthetic efficiencies of H₂O₂ in three separate ionic salts (NaCl, MgCl₂, and CaCl₂) were plotted in Figure 4g. For this study, we controlled the concentration of salt ions around their respective concentrations in seawater, according to the international standard of seawater composition³⁴. It should be noted that, in the range of seawater

concentration, the addition of NaCl, MgCl₂ and CaCl₂ improve the H₂O₂ yield of the photocatalyst.

Comment 2:

In Figure 4g contradicts with suggested role of Na⁺, Ca²⁺, Mg²⁺ in seawater as promoters for electron extraction in photocatalysis involving CDs. The current mechanistic discussion behind the seawater effect is very speculative (line 98-118, and line 317-322) and should be improved by theoretical calculation and considerations. Additionally, it is known that high salt concentrations/ionic strength can destabilize nanocatalysts by aggregation as a result of electric double layer suppression. There is no such consideration and the enhanced H₂O₂ formation in seawater appears to disagree with catalyst aggregation in seawater. There are also important considerations in the revision of the manuscript.

Response 2:

Thank you for your valuable suggestions.

In Figure 4g, although hydrogen peroxide production rate shows a slight decrease in 0.1 M NaCl solution, it was within the error range. For salt solutions (Na⁺, Ca²⁺, Mg²⁺), the production rate of hydrogen peroxide increase with the increase of salt concentration in general. So, the Na⁺, Ca²⁺, Mg²⁺ in seawater can be used as promoters for electron extraction in photocatalysis involving CDs.

The theoretical calculation has been performed and provided to verify our views. The DFT calculation results (Figure S3) of CDs with and without Na⁺ show that the addition of salt ions can increase the electron sink of CDs.

Furthermore, salting out experiments were carried out in ultrapure water, seawater and high concentration NaCl solution (2.5 and 5 mol L⁻¹). Figure R2 show the catalyst dispersion in salt solution (5 M NaCl) at different standing times, suggesting that the catalyst could aggregate at high concentration salt solution. Besides, the experimental results in Figure S19 show that although the catalyst aggregate when the sodium chloride concentration is too high, it does not appear in the range of seawater

concentration.

The revised parts are shown as follows:

The full X-ray photoelectron spectroscopy (XPS) spectrum in Figure S2a demonstrates only C and O elements in CDs. The high-resolution spectrum of C 1s can be fitted for C-C/C=C, C-O and C=O while the O 1s can be matched to O_{surf} and O_{ads}^{26,27}. A further structure analysis on CDs is shown in Figure 1c, where the hydron in carboxyl group shows electropositivity (δ_1^+) while the adjacent oxygen is electronegativity (δ_1^-). When the metal salt (M^+) is added, the carboxyl group can be ionized, and the electronegativity (δ_1^-) of oxygen in carbon group increase, which will, in turn, increase the charges and electron sink barrier of CDs, and further prolongs the life of the electron. Density functional theory (DFT) calculations have been conducted to understand the effect of ions on the electrostatic potentials of CDs with respect to the energy level of a vacuum. As shown in inset images of Figure S3, CDs were modelled with one dimensional graphene nanoribbons, the edge of which were functionalized with carboxyl (-COOH) and sodium carboxylate (-COONa). The computational details can be found in the supporting information. The work function of these graphene nanoribbons can be estimated by plane-averaged potentials. Figure S3 shows the plane-averaged potentials of nanoribbons are plotted along the direction that perpendicular to the surface. It is seen that the trapping of electrons is more profound in the graphene nanoribbon terminated with carboxyl groups. This is in good agreement with our predictions.

1.8 The salting out experiments

10 mg PM-CDs-30 was added to ultrapure water (50mL), seawater (50mL), 2.5 mol L⁻¹ NaCl solution (50mL) and 5 mol L⁻¹ NaCl solution (50mL). The mixture was then ultrasonic for 10 min to disperse evenly. Next, the mixed solution was left to stand, and the upper solution was taken at intervals (0, 1, 1, 5, 8, 12, and 24 h) for UV-vis test to determine the content of catalyst in the solution. The results are shown in Figure S19.

2. Supplemental computational details and methods

The Vienna Ab initio Simulation Package (VASP) were applied to perform the spin-polarized density functional theory (DFT) calculations. The electron-ion interactions were described by the projector augmented wave (PAW) method proposed by Blöchl and implemented by Kresse. The electronic ground states were treated with the Perdew-Burke-Ernzerhof (PBE) within the generalized gradient approximation (GGA) exchange correlations potentials. The cut-off energy of plane wave basis was set as 400 eV, and the van der Waals interactions were described with the vdW-D3 method. The graphene ribbon was modeled with a 4×4 unit cell in the type of zigzag in which including 32 carbon atoms. For the Brillouin zone sampling, a 3×3 K point mesh was used. Vacuum region of 22 Å and 36 Å were applied separately along two directions to avoid the interactions between transnationally periodic images. During the structure optimization, all the atoms in the cell were allowed to relax. The optimization was stopped when the force residue on the atom was smaller than 0.02 eV/Å. The climbing image nudged elastic band (CI-NEB) method with six images was applied for searching the minimum energy paths of all reactions and finding the transition states. The transition states were then picked as the input structures of the subsequent dimer calculations.

Figure S3. The plane-averaged potentials of nanoribbons plotted along the direction

that perpendicular to the surface. The atomic configurations of carboxyl (-COOH) and sodium carboxylate (-COONa) graphene nanoribbons are shown in the inset image.

Figure S19. Changes in catalyst (PM-CDs-30) dispersion concentration over time in different solutions.

Figure R2. Diagram of catalyst dispersion in salt solution (5 M NaCl) at different standing times. (a) 0 h. (b) 10 h.

Comment 3:

The performance of photocatalyst in seawater should be compared to those in existing work using seawater, to explicitly highlight uniqueness of this work. The AQYs can also be compared among related photocatalytic H_2O_2 generation studies.

Response 3:

Thank you for your valuable suggestions.

The composite catalyst we prepared can not only produce hydrogen peroxide in water, but also effectively promote the production of hydrogen peroxide by using salt ions in seawater. This is a major breakthrough in the photocatalytic production of hydrogen peroxide. There are only a few works on photocatalytic production of hydrogen peroxide in seawater. Through comparison the catalytic activities of different photocatalysts systems in water and seawater in Table S4, we found that the catalyst we prepared has great advantage in the production of hydrogen peroxide by photocatalysis in seawater.

The revised parts are shown as follows:

Table S4. Comparison of the catalytic activities of different photocatalyst systems in the literatures.

Photocatalyst	Condition	H ₂ O ₂ μmol h ⁻¹ g ⁻¹	AQY at 420 nm	SCC %	Ref.
PM-CDs-30	Real seawater	1776	0.99	0.21	This work
m-WO ₃ /FTO		2			
photoanode-Co ^{II} (Ch)/CP cathode	Artificial seawater	mmol L ⁻¹ h ⁻¹	-	0.55	5
TiO ₂	4% NaCl solution	90	-	-	6
FeO(OH)/BiVO ₄ /FTO		6.5			
photoanode-Co ^{II} (Ch)/carb on paper cathode	Artificial seawater	mmol L ⁻¹ h ⁻¹	-	0.89	7
Au/BiVO ₄	O ₂ -saturated water	2.412	0.24%	-	8
g-C ₃ N ₄ /NaBH ₄	Water	170	4.3%	0.26	9
g-C ₃ N ₄ /BDI	O ₂ -saturated water	17	2.6%	0.13	10
g-C ₃ N ₄ /PDI/rGO	O ₂ -saturated water	23.4	6.1%	0.2	11

C ₃ N ₄ / anthraquinone	O ₂ -saturated water	361	4.8%	0.178	3
RF523	O ₂ -saturated water	82.5	6%	0.5	12

Supplemental References

- Mase, K., Yoneda, M., Yamada, Y. & Fukuzumi, S. Seawater usable for production and consumption of hydrogen peroxide as a solar fuel. *Nat. Commun.* **7**, 11470 (2016).
- Harada, H. Isolation of hydrogen from water and/or artificial seawater by sonophotocatalysis using alternating irradiation method. *Int. J. Hydrogen Energ.* **26**, 303-307 (2001).
- Mase, K., Yoneda, M., Yamada, Y. & Fukuzumi, S. Efficient photocatalytic production of hydrogen peroxide from water and dioxygen with bismuth vanadate and a cobalt(II) chlorin complex. *ACS Energy Lett.* **1**, 913–919 (2016).
- Hirakawa, H. *et al.* Au nanoparticles supported on BiVO₄: effective inorganic photocatalysts for H₂O₂ production from water and O₂ under visible light. *ACS Catal.* **6**, 4976–4982 (2016).
- Zhu, Z., Pan, H., Muruganathan, M., Gong, J. & Zhang, Y. Visible light-driven photocatalytically active g-C₃N₄ material for enhanced generation of H₂O₂. *Appl. Catal. B-Environ.* **232**, 19–25 (2018).
- Kofuji, Y. *et al.* Graphitic carbon nitride doped with biphenyl diimide: efficient photocatalyst for hydrogen peroxide production from water and molecular oxygen by sunlight. *ACS Catal.* **6**, 7021–7029 (2016).
- Kofuji, Y. *et al.* Carbon nitride–aromatic diimide–graphene nanohybrids: metal-free photocatalysts for solar-to-hydrogen peroxide energy conversion with 0.2% Efficiency. *J. Am. Chem. Soc.* **138**, 10019–10025 (2016).
- Shiraishi, Y. *et al.* Resorcinol–formaldehyde resins as metal-free semiconductor photocatalysts for solar-to-hydrogen peroxide energy conversion. *Nat. Mater.* **18**, 985–993 (2019).

Comment 4:

It is suggested that H₂O acts the electron donor and forms O₂ (i.e., organic donor free). The authors should consider to measure O₂ evolution to confirm this important claim.

Response 4:

Thanks to your valuable comments.

The change in oxygen production over time has been shown in Figure S23, suggesting that the production of oxygen is a process of continuous increase over time. In other words, during the reaction, H₂O acts as the electron donor and forms O₂.

The added Figure is shown as follows.

Figure S23. Amounts of O₂ and H₂O₂ formed during the half photoreaction. Considerations: water (20 mL), PM-CDs-30 catalyst (10 mg), AgNO₃ (10 mM), N₂.

Comment 5:

The photocatalyst PM-CDs-30 actually shows some oxidation after photocatalysis as in Figure S17b. The relative C-O group fraction increases as compared to that in the parent photocatalyst (Figure 2b). Could this oxidation provide the electron for O₂ reduction and H₂O₂ formation? The measurement of O₂ formation (H₂O as the electron donor) should be quantitatively compared to that of H₂O₂ formation, so that other electron sources can be evaluated and establish H₂O as the electron donor in the system.

Response 5:

Thank you for your precise suggestions.

Figure S2 is the XPS spectra of CDs. The XPS spectra of PM-CDs-30 before and after reaction are shown in Figure S10 and Figure S21, respectively. It can be seen from the figures that the peak area of C-O bond does not increase, indicating that the catalyst was not oxidized. Therefore, the photocatalyst PM-CDs-30 was not oxidized to provide electrons for O₂ reduction and H₂O₂ formation.

The evolution of oxygen over time (the half photoreaction) in Figure S23 indicates a continuous linear increase in oxygen, indicating that catalyst has not been oxidized or deteriorated during the photocatalytic reaction.

In seawater (or water) and N₂-saturated seawater (or N₂-saturated water), the catalyst did not catalyze the reaction to produce hydrogen peroxide under dark condition (Figure S24), indicating that the catalyst was not oxidized by oxygen in the air to provide electrons, and there was no other source of electron donor in the catalytic reaction system. In N₂-saturated seawater, the hydrogen peroxide yield under light irradiation is 25 μmol (16 μmol for N₂-saturated water). Here, the hydrogen peroxide comes from the oxygen produced by the oxidation of water, which is reduced to hydrogen peroxide by catalyst. After photocatalytic reaction, the sample was characterized by XPS again, and the result is shown in Figure R3. Similar to the above result, the C-O bond does not increase after the catalytic reaction, indicating that the catalyst was not oxidized to provide electrons. Therefore, in the whole photocatalytic system, the only driving forces of the catalytic reaction are photogenerated electrons and holes. These photogenerated carries further undergo redox reactions in the catalyst system.

The revised parts are shown as follows:

Figure S2. XPS spectra of CDs. (a) Full spectrum. (b) C 1s spectrum. (c) O 1s spectrum.

Figure S10. XPS spectra of PM-CDs-30. (a) Full spectrum. (b) C 1s spectrum. (c) O 1s spectrum.

Figure S21. XPS spectra of PM-CDs-30 after stability tests. (a) Full spectrum. (b) C 1s spectrum. (c) O 1s spectrum.

Figure S23. Amounts of O₂ and H₂O₂ formed during the half photoreaction.
Considerations: water (20 mL), PM-CDs-30 catalyst (10 mg), AgNO₃ (10 mM), N₂.

Figure S24. The production of H₂O₂ by photocatalytic reaction in different conditions.

Figure R3. C 1s spectra of PM-CDs-30 after reaction in seawater.

Comment 6:

Line 242 the suggested H₂O₂ decomposition should be evaluated/measured in the presence of photocatalysts with added H₂O₂.

Response 6:

Thank you for your precise suggestions.

The decomposition experiment of hydrogen peroxide has been carried out, and the results shown in Figure S18 suggests that the catalyst PM-CDs-30 has low ability of hydrogen peroxide decomposition.

The revised parts are shown as follows:

1.6 Electrochemical measurement of the PM-CDs-30 for decomposition of H₂O₂

The H₂O₂ decomposition behavior of PM-CDs-30 was measured by cycle voltammetry (CV) in 0.2 M (pH = 7) phosphate buffered solution with 25 mM H₂O₂. A standard three-electrode system with CHI 760E workstation was used. The carbon electrode and the saturated calomel electrode were used as the counter electrode and the reference electrode, respectively. The bare glassy carbon (GC) electrode (3 mm diameter) and PM-CDs-30-modified GC were used as the work electrodes, which are used as the control group and experiment group respectively. Here, 4 μ L catalyst solution (2 mg mL⁻¹) and 5 μ L of 0.5 wt % Nafion solution were dropped onto the working area of a cleaned GC electrode and put naturally to dry. The CV curves were measured under darkness with a scan rate of 50 mV s⁻¹ and the results are shown in Figure S18a.

1.7 Degradation of H₂O₂ by PM-CDs-30

10 mg PM-CDs-30 was dispersed in 20 mL water with 50 μ mol H₂O₂ added. The reaction vessels were then placed under dark conditions and stirred. The content of H₂O₂ in the solution was detected after 0, 6, 12, and 24h. The change of H₂O₂ content is shown in Figure S18b.

Figure S18. (a) Cyclic voltammograms curves of bare GC and PM-CDs-30 modified GC electrodes in 0.2 M (pH=7) phosphate buffered 25 mM H₂O₂ solution at a scan

rate of 50 mV s^{-1} (without light). (b) Change of H_2O_2 content in solution over time under dark condition.

Comment 7:

Line 255 considering the amount of H_2O_2 formed $>1000 \text{ umole}$ (Figure 3f), the required O_2 concentration would exceed 50 mM (20 mL water) in water. How could this be possible given the solubility limit of O_2 ? Did the experiment actively bubble air/ O_2 in the solution? Please clarify.

Response 7:

Thank you for your valuable comment. In a typical photocatalytic reaction, the catalytic reaction system is an open system to ensure sufficient oxygen to participate in the reaction. Although the dissolved oxygen in 20 mL water cannot meet the needs of hydrogen peroxide production, with the consumption of oxygen in water, the oxygen in the air would continue to diffuse into the catalytic reaction system, which provides a guarantee for the catalytic reaction. In addition, in this photocatalytic reaction system, photocatalytic water decomposition would produce oxygen, providing a certain basis for oxygen reduction reaction.

The detailed operation of catalytic reaction has been supplemented in the experimental part, and the revised part is shown as follows:

Experiment of photocatalysis

The photocatalytic activity of the photocatalysts was assessed by a multichannel photochemical reaction system (PCX-50B, Beijing Perfectlight Co. Ltd, China) equipped with a visible light source ($\lambda \geq 420 \text{ nm}$), without the addition of sacrificial agents and other cocatalysts. In a typical reaction, 10 mg of the catalyst was dispersed in a quartz bottle (50 mL) containing 20 mL of seawater and then it was exposed to visible light for 5 h without seal in air at room temperature. The rotation speed was 180 rpm. The optimal activity and reaction mechanism of the catalyst were explored by changing the light condition (with and without light), the illumination time (5, 12, 24, 48, 72, 84 and 96 h), reaction environment (water or seawater), the excitation wavelength ($\lambda = 365, 420, 535$ and 620 nm) and the sacrificial agent (AgNO_3 ,

EDTA-2Na, tert-butyl alcohol (TBA) and benzoquinone (BQ)) (2 mM). After each reaction, the catalyst was washed and dried for stability testing (5 h for each cycle).

Experiments on the influence of rotating speed on catalytic activity were carried out at different rotating speeds, which were 40, 80, 120, 160 and 200 rpm, respectively. Here, 10 mg catalyst was dispersed in 20 mL seawater (or pure water), and the quartz bottle (50 mL) was not sealed. The catalytic reaction took place in air at room temperature under visible light irradiation ($\lambda \geq 420$ nm) for 5 h. For the experiment of the relationship between oxygen partial pressure and catalytic activity, the catalytic system was sealed, and then oxygen (0, 4, 8, 12 and 16 mL) was injected into it to control the oxygen partial pressure of the catalytic system. Then, the catalytic reaction was carried out at the 180 rpm under visible light irradiation ($\lambda \geq 420$ nm) for 5 h.

For the reaction atmosphere, N_2 (or O_2) was used to bubble in seawater for 20 min, and then the catalytic activity of the catalyst was tested under different atmospheres under airtight condition. As for the effect of different types of salts on the activity, different concentrations of NaCl solution, $MgCl_2$ solution and $CaCl_2$ solution were employed (the concentrations were within the range of seawater concentration).

In order to detect the change of hydrogen peroxide yield with temperature, 20 mL seawater containing 10 mg PM-CDs-30 was performed at different temperatures (20, 40, 60 and 80 °C) under visible light irradiation for 5 h. In addition, the temperature change of the reaction solution (20 mL seawater with 10 mg catalyst) during the photocatalytic reaction was monitored by an infrared camera under room temperature and unsealed conditions.

To monitor the evolution of O_2 , 10 mg of the catalyst was dispersed in a quartz bottle (50 mL) containing 20 mL of water and then 10 mM of AgCl was added. The system was bubbled by N_2 for 30 min. The sealed reaction system reacted 2, 4 and 6 h under light irradiation, and then gas chromatography was used to detect the change of oxygen.

At the end of the experiment, the solution was centrifuged, filtered, and then the yield of H_2O_2 was determined by 0.1 M $KMnO_4$.

Comment 8:

CD is suggested to act as the electron trap. To confirm this, the photoluminescence indicative of electron-hole re-combination should be measured for the various photocatalysts made.

Response 8:

Thank you for your valuable suggestions.

The photoluminescence of the various photocatalysts has been measured and provided in Figure R4. As shown in the Figure R4, at the excitation wavelength of 350 nm, neither the pure polymer (PM-CDs-0) nor the carbon-based composite catalyst (PM-CDs-30 and PM-CDs-100) has an emission peak, which means that the catalysts have no photoluminescence property. Therefore, the photoluminescence measurement cannot characterize the electron-hole recombination in this catalytic system. So, we used transient photovoltage and other photoelectric measurements to detect the properties and reaction mechanism of the catalysts.

The PL spectra are shown as follows:

Figure R4. Photoluminescence spectra of PM-CDs-0, PM-CDs-30 and PM-CDs-100.

Comment 9:

The Figure labels (including x, y axis labels) especially Figure 3&4 are too small to read. The sub-figures in Figure 3 are mis-labeled in the caption and Figure 3b&c are not introduced in the caption. Typos in the Figure 3 labels should be corrected. In

Figure 3d&e, the model calculation should be expressed as lines and measurement data in points. Additionally, more description of the experimental conditions used in Figure 3&4 is useful, as the photocatalysis experiments described in method section is very generally, for example, not covering how rotation and oxygen partial pressure were conducted.

Response 9:

Thank you for your valuable suggestions. We have revised the Figure labels in Figure 3 and 4, and provided a more detailed description of the photocatalytic experiments.

The revised parts are shown as follows:

Figure 3. Catalytic performance of PM-CDs-30. (a) Comparison of H_2O_2 production among the photocatalysts with different CDs contents in pure water and real seawater. (b) Comparison of hydrogen peroxide yield rate between seawater and water with different catalysts. (c) The hydrogen peroxide production ratio with different CDs contents to pure polymer catalyst PM-CDs-0. (d) Calculation and

experiment on the dependence of H_2O_2 production rate of PM-CDs-30 and rotational speed in different medium (red line: calculation date; black point: experiment date). (e) Calculation and experiment on the dependence of H_2O_2 production rate of PM-CDs-30 and oxygen partial pressure in different medium (red line: calculation date; black point: experiment date). (f) Time course of H_2O_2 photoproduction by PM-CDs-30. (g) Stability of photoproduction of H_2O_2 by PM-CDs-30. (h) Wavelength-dependent AQY of oxygen reduction reaction by PM-CDs-30.

Figure 4. Photocatalytic mechanism of PM-CDs-30. TPV curves of (a) PM-CDs-0, (b) PM-CDs-30 before and after adding NaCl. (c) The photocatalytic H_2O_2 production rate upon the addition of different sacrificial agents. (d) EPR diagrams of PM-CDs-30 under darkness and light. (e) The production rate of H_2O_2 by photocatalytic reaction in different gas environments. (f) TPV curves of PM-CDs-30 under different

conditions. (g) H_2O_2 production rate varies with ion concentration in different salt solutions.

Experiment of photocatalysis

The photocatalytic activity of the photocatalysts was assessed by a multichannel photochemical reaction system (PCX-50B, Beijing Perfectlight Co. Ltd, China) equipped with a visible light source ($\lambda \geq 420$ nm), without the addition of sacrificial agents and other cocatalysts. In a typical reaction, 10 mg of the catalyst was dispersed in a quartz bottle (50 mL) containing 20 mL of seawater and then it was exposed to visible light for 5 h without seal in air at room temperature. The rotation speed was 180 rpm. The optimal activity and reaction mechanism of the catalyst were explored by changing the light condition (with and without light), the illumination time (5, 12, 24, 48, 72, 84 and 96 h), reaction environment (water or seawater), the excitation wavelength ($\lambda = 365, 420, 535$ and 620 nm) and the sacrificial agent (AgNO_3 , EDTA-2Na, tert-butyl alcohol (TBA) and benzoquinone (BQ)) (2 mM). After each reaction, the catalyst was washed and dried for stability testing (5 h for each cycle).

Experiments on the influence of rotating speed on catalytic activity were carried out at different rotating speeds, which were 40, 80, 120, 160 and 200 rpm, respectively. Here, 10 mg catalyst was dispersed in 20 mL seawater (or pure water), and the quartz bottle (50 mL) was not sealed. The catalytic reaction took places in air at room temperature under visible light irradiation ($\lambda \geq 420$ nm) for 5 h. For the experiment of the relationship between oxygen partial pressure and catalytic activity, the catalytic system was sealed, and then oxygen (0, 4, 8, 12 and 16 mL) was injected into it to control the oxygen partial pressure of the catalytic system. Then, the catalytic reaction was carried out at the 180 rpm under visible light irradiation ($\lambda \geq 420$ nm) for 5 h.

For the reaction atmosphere, N_2 (or O_2) was used to bubble in seawater for 20 min, and then the catalytic activity of the catalyst was tested under different atmospheres under airtight condition. As for the effect of different types of salts on the activity, different concentrations of NaCl solution, MgCl_2 solution and CaCl_2 solution were employed (the concentrations were within the range of seawater concentration).

In order to detect the change of hydrogen peroxide yield with temperature, 20 mL seawater containing 10 mg PM-CDs-30 was performed at different temperatures (20, 40, 60 and 80) under visible light irradiation for 5 h. In addition, the temperature change of the reaction solution (20 mL seawater with 10 mg catalyst) during the photocatalytic reaction was monitored by an infrared camera under room temperature and unsealed conditions.

To monitor the evolution of O₂, 10 mg of the catalyst was dispersed in a quartz bottle (50 mL) containing 20 mL of water and then 10 mM of AgCl was added. The system was bubbled by N₂ for 30 min. The sealed reaction system reacted 2, 4 and 6 h under light irradiation, and then gas chromatography was used to detect the change of oxygen.

At the end of the experiment, the solution was centrifuged, filtered, and then the yield of H₂O₂ was determined by 0.1 M KMnO₄.

Comment 10:

There are many errors (grammars, typos, mistakes) in current text that should be carefully corrected.

Response 10:

Thank you for your valuable suggestion. We have checked the English grammar and corrected the typos in the text.

REVIEWER COMMENTS

Reviewer #1 (Remarks to the Author):

The authors carefully revised the manuscript regarding the problems. All my questions have been addressed. Therefore, I recommend it to be published as is.

Reviewer #2 (Remarks to the Author):

The authors have well addressed all the questions and performed the additional experiments required. The manuscript has been improved and can be accepted in the present form.

Reviewer #3 (Remarks to the Author):

Some of my first review questions are addressed, but some still remain as outlined as follows.

1. I am not convinced that the addition of salts, particularly Na^+ and Mg^{2+} , significantly enhances H_2O_2 photoproduction (Figure 4g), given the large error bars of measurements at various salt concentrations. I would suggest more rigorous statistical analysis to support their claims.

2. Table S4: the performance of the photocatalyst reported in this work should be explicitly discussed in comparison to those in existing work using seawater. It is not clear whether the performance of this work is actually better. The SCC of this work is not better (0.21% vs 0.55% and 0.89% in ref 5 and ref 7), and the H_2O_2 production rates of the literature work are shown as different units, which are not easy for direct comparison. Table S4: The AQY of this work is indicated as 0.99. It is 0.99 or 0.99%?

3. The relative contents of O 1s and C 1s in the XPS survey spectra before (Figure S10a) and after (Figure S21a) photocatalysis have changed with the one after photocatalysis showing strongly increased O1s. Does this indicate that the photocatalyst has oxidized? Also while the C-O concentration does not increase, the C=O concentration increases after photocatalysis (Figure S21 b). Please provide the overall quantitative oxygen functionalities concentrations including C-O and C=O before and after photoreaction for a definitive comparison.

Other comments:

Figure S19: % in the y-axis label should be removed.

Figure 3 caption: calculation date and experiment date should be corrected to calculation data and experimental data.

The English language should again be generally improved before publication.

Responses to the Referees' Comments

we sincerely thank for the referees' work on our manuscript. Herein, we respond to the referees' insightful comments and suggestions in detail.

Referee: #1

The authors carefully revised the manuscript regarding the problems. All my questions have been addressed. Therefore, I recommend it to be published as is.

Response:

Thank you for your recognition and guidance of our work.

Referee: #2

The authors have well addressed all the questions and performed the additional experiments required. The manuscript has been improved and can be accepted in the present form.

Response:

Thank you for your recognition and guidance of our work.

Referee: #3

Comments to the Author

Some of my first review questions are addressed, but some still remain as outlined as follows.

Comment 1:

I am not convinced that the addition of salts, particularly Na^+ and Mg^{2+} , significantly enhances H_2O_2 photoproduction (Figure 4g), given the large error bars of measurements at various salt concentrations. I would suggest more rigorous statistical analysis to support their claims.

Response 1:

Thank you for your precise viewpoint. The statistical method was adopted to test the difference significance between the experimental results of ion concentration, with the results showing in Figure 4g. According to statistical analysis, there is significant difference between the experimental results of pure water and high concentration NaCl solution (similar concentration in seawater, 0.5 mol/L), while there is no significant difference between the experimental results in low concentration NaCl solutions. The mainly reason is that the promoting effect on hydrogen peroxide production is not obvious, when the concentration of Na^+ is very low. Similarly, the experimental results between pure water and high concentration MgCl_2 solutions (similar concentration in seawater, ≥ 0.04 mol/L) are also significantly different. And there are significant differences between the experimental results of pure water and CaCl_2 solution. Based on the above analysis, when the concentration of salt ions is the range of seawater, there is a significant difference between the experimental results and that of pure water, indicating that the addition of salts can promote the production of hydrogen peroxide.

The revised parts are shown as follows:

Figure 4. Photocatalytic mechanism of PM-CDs-30. TPV curves of (a) PM-CDs-0, (b) PM-CDs-30 before and after adding NaCl. (c) The photocatalytic H₂O₂ production rates upon the addition of different sacrificial agents. (d) EPR diagrams of PM-CDs-30 under darkness and light. (e) The production rate of H₂O₂ by photocatalytic reaction in different gas environments. (f) TPV curves of PM-CDs-30 under different conditions. (g) H₂O₂ production rate varies with ion concentration in different salt solutions. Single asterisks indicate $P < 0.10$; double asterisks indicate $P < 0.05$; triple asterisks indicate $P < 0.01$.

In the presence of CDs, the TPV curves of PM-CDs-30 mixed with other salt ions (Figure S25), such as MgCl₂ and CaCl₂, also show similar phenomenon to NaCl. In addition, the difference significance test of statistics method was adopted in the final analysis, indicating that the cations can promote the production of hydrogen peroxide in seawater.

Comment 2:

Table S4: the performance of the photocatalyst reported in this work should be explicitly discussed in comparison to those in existing work using seawater. It is not clear whether the performance of this work is actually better. The SCC of this work is not better (0.21% vs 0.55% and 0.89% in ref 5 and ref 7), and the H₂O₂ production rates of the literature work are shown as different units, which are not easy for direct comparison. Table S4: The AQY of this work is indicated as 0.99. It is 0.99 or 0.99%?

Response 2:

Thank you for your valuable suggestions. In the existing works, there are a few works on the production of hydrogen peroxide through catalysis in seawater. So far, the production of hydrogen peroxide in seawater is all accomplished by electrocatalysis or photoelectrocatalysis. In particular, the photocatalytic production of hydrogen peroxide by metal-free catalyst in seawater has not been reported. In this work, therefore, the photocatalytic production of hydrogen peroxide using metal-free catalyst in seawater is a breakthrough.

For the high SCC in seawater mentioned in ref. 5 and ref. 7, the catalysts used in these two works are metal oxides, using the photoelectrocatalytic method. In addition, the catalytic reactions in these literatures were carried out in the O₂ saturated acidic seawater containing HClO₄ and NaClO₄. The reaction conditions in our work are only catalyst and seawater, with no other additives. We use the natural characteristics of seawater to improve the catalytic efficiency, which is of great significance for the utilization of seawater.

Finally, we revised the efficiency in Table S4 to a uniform unit and corrected the AQY of 0.99 to 0.99%. The revised parts are shown as follows:

Table S4. Comparison of the catalytic activities of different photocatalyst systems in the literatures.

Photocatalyst	Condition	H ₂ O ₂	AQY	SCC	Ref.
---------------	-----------	-------------------------------	-----	-----	------

		$\mu\text{mol/h}$	at 420 nm	%	
PM-CDs-30	Real seawater	17.76	0.99%	0.21	This work
m-WO₃/FTO-Co^{II}(Ch)/CP cathode	O ₂ -saturated artificial seawater	16	-	0.55	5
TiO₂	HClO ₄ (pH=1.3) and 0.1 M NaClO ₄	18	-	-	6
FeO(OH)/BiVO₄/FTO photoanode-Co^{II}(Ch)/carbon paper cathode	O ₂ -saturated artificial seawater	34	-	0.89	7
Au/BiVO₄	HClO ₄ (pH=1.3) and 0.1 M NaClO ₄	0.121	0.24%	-	8
g-C₃N₄/NaBH₄	O ₂ -saturated water	17	4.3%	0.26	9
g-C₃N₄/BDI₅₀	Water	0.854	2.6%	0.13	10
g-C₃N₄/PDI/rGO	O ₂ -saturated water	1.21	6.1%	0.2	11
RF523	O ₂ -saturated water	2.58	6%	0.5	12

Comment 3:

The relative contents of O 1s and C 1s in the XPS survey spectra before (Figure S10a) and after (Figure S21a) photocatalysis have changed with the one after photocatalysis showing strongly increased O1s. Does this indicate that the photocatalyst has oxidized? Also while the C-O concentration does not increase, the C=O concentration increases after photocatalysis (Figure S21 b). Please provide the overall quantitative oxygen functionalities concentrations including C-O and C=O before and after photoreaction for a definitive comparison.

Response 3:

Thank you for your valuable suggestions. In the XPS survey spectra before and after

the photocatalysis, the intensity of O 1s increases, which is due the absorption of water and oxygen on the surface of the catalyst. In addition, the overall oxygen functionalities concentrations were quantitated (Table S5). The results show that there is no significant different in the overall oxygen functionalities concentrations in the catalyst before and after the photocatalytic reaction, indicating that the catalyst was not oxidized.

Table S5. The carbon and oxygen functionalities concentrations in C 1s spectra.

	Before reaction	After reaction
	%	%
C-C	60.6	61.6
C-O/C=O	39.4	38.4

Comment 4:

Figure S19: % in the y-axis label should be removed.

Response 4:

Thanks to your valuable comments. We have removed the % in the y-axis label, and the revised Figure is shown as follows.

Figure S19. Changes in catalyst (PM-CDs-30) dispersion concentration over time in different solutions.

Comment 5:

Figure 3 caption: calculation date and experiment date should be corrected to calculation data and experimental data.

Response 5:

Thank you for your precise suggestions. We have corrected calculation date and experiment date to calculation data and experimental data. The revised part is shown as follows:

Figure 3. Catalytic performance of PM-CDs-30. (a) Comparison of H₂O₂ production among the photocatalysts with different CDs contents in pure water and real seawater. (b) Comparison of hydrogen peroxide yield rate between seawater and water with different catalysts. (c) The hydrogen peroxide production ratio with different CDs contents to pure polymer catalyst PM-CDs-0. (d) Calculation and experimental on the dependence of H₂O₂ production rate of PM-CDs-30 and rotational speed in different medium (red line: calculation data; black point: experimental data). (e) Calculation and experimental on the dependence of H₂O₂ production rate of PM-CDs-30 and oxygen partial pressure in different medium (red line: calculation data; black point: experimental data). (f) Time course of H₂O₂ photoproduction by PM-CDs-30. (g) Stability of photoproduction of H₂O₂ by PM-CDs-30. (h) Wavelength-dependent AQY of oxygen reduction reaction by PM-CDs-30.

Comment 6:

The English language should again be generally improved before publication.

Response 6:

Thank you for your valuable suggestion. We have polished the whole paper to avoid apparent grammar mistakes.

REVIEWERS' COMMENTS

Reviewer #3 (Remarks to the Author):

The authors have satisfactorily addressed my comments/questions.

Response to the Reviewer's Comments

We sincerely thank for the referees' work on our manuscript. Herein, we respond to the referees' insightful comments and suggestions in detail.

Referee: #3

The authors have satisfactorily addressed my comments/questions.

Response:

Thank you for your recognition and guidance of our work.